*Report*

# Foxp1 suppresses cortical angiogenesis and attenuates HIF-1alpha signaling to promote neural progenitor cell maintenance

Jessie E Buth[1,5], Catherine E Dyevich[2,5], Alexandra Rubin[2], Chengbing Wang [2], Lei Gao [3], Tessa Marks[3], Michael RM Harrison[4], Jennifer H Kong [3], M Elizabeth Ross [2], Bennett G Novitch [1,6] & Caroline Alayne Pearson [2,6✉]

## Abstract

**Neural progenitor cells within the cerebral cortex undergo a characteristic switch between symmetric self-renewing cell divisions early in development and asymmetric neurogenic divisions later. Yet, the mechanisms controlling this transition remain unclear. Previous work has shown that early but not late neural progenitor cells (NPCs) endogenously express the autism-linked transcription factor Foxp1, and both loss and gain of Foxp1 function can alter NPC activity and fate choices. Here, we show that premature loss of Foxp1 upregulates transcriptional programs regulating angiogenesis, glycolysis, and cellular responses to hypoxia. These changes coincide with a premature destabilization of HIF-1α, an elevation in HIF-1α target genes, including Vegfa in NPCs, and precocious vascular network development. In vitro experiments demonstrate that stabilization of HIF-1α in Foxp1-deficient NPCs rescues the premature differentiation phenotype and restores NPC maintenance. Our data indicate that the endogenous decline in Foxp1 expression activates the HIF-1α transcriptional program leading to changes in the tissue environment adjacent to NPCs, which, in turn, might alter their self-renewal and neurogenic capacities.**

**Keywords** Corticogenesis; Angiogenesis; HIF-1 Signaling; Autism; Neurodevelopment

**Subject Categories** Development; Neuroscience; Vascular Biology & Angiogenesis

## Introduction

The temporal specification of neuronal subtypes in the mammalian cortex arises through progressive changes in the transcriptional states of cortical progenitors termed radial glia (RG) (Desai and McConnell, 2000; Shen et al, 2006). Early in development, RG undergo symmetric self-renewing cell divisions to expand the progenitor pool. Early RG (~embryonic day E12–13 in mouse) have the potential to generate all classes of excitatory glutamatergic neurons and glia (Beattie and Hippenmeyer, 2017). Late RG (~E13.5–15.5 in mice) generate basal progenitors, including intermediate progenitors and basal radial glia that, in turn, amplify the generation of later-born excitatory neurons which make up the upper layers of the cortex (Taverna et al, 2014). The precise timing of the switch from multipotent RG to more restricted RG is integral to the layered organization of the cortex and orderly assembly of circuits.

We previously demonstrated that early RG highly express the autism-linked transcription factor Foxp1, which promotes symmetric cell divisions and self-renewal, and sustains the potential to generate both early-born deep-layer neurons and later-born upper-layer neurons (Pearson et al, 2020). The endogenous down-regulation of Foxp1 in late RG at mid-neurogenic stages is required to transition to asymmetric neurogenic divisions. Conditional removal of Foxp1 function from early RG resulted in a premature transition to intermediate progenitor generation and neuronal differentiation, resulting in a reduction of early-born deep-layer neurons and a concomitant increase in upper-layer neurons (Pearson et al, 2020). However, the mechanisms through which Foxp1 promotes early RG character were unclear.

Recent single-cell RNA-Seq studies have shown that RG become increasingly responsive to extrinsic signals in the embryonic environment as development proceeds, particularly metabolic substrates made available through the developing vascular network (Telley et al, 2019; Dong et al, 2022). The switch of RG from symmetric self-renewing cell divisions to asymmetric neurogenic divisions has been associated with the relief from hypoxia as the vascular network forms and changes in RG glycolytic activity (Lange et al, 2016; Komabayashi-Suzuki et al, 2019).

Here, we show that early loss of Foxp1 in RG leads to transcriptional changes in genes associated with angiogenesis,

[1]Department of Neurobiology, Eli and Edythe Broad Center of Regenerative Medicine and Stem Cell Research, Intellectual and Developmental Disabilities Research Center, David Geffen School of Medicine at UCLA, Los Angeles, CA 90095, USA. [2]Feil Family Brain and Mind Research Institute and Center for Neurogenetics, Weill Cornell Medicine, New York, NY 10021, USA. [3]Department of Biochemistry, University of Washington, Seattle, WA 98195, USA. [4]Cardiovascular Research Institute, Weill Cornell Medicine, New York, NY 10021, USA. [5]These authors contributed equally: Jessie E Buth, Catherine E Dyevich. [6]These authors contributed equally as senior authors: Bennett G Novitch, Caroline Alayne Pearson. ✉E-mail: cap4010@med.cornell.edu

Hypoxia Inducible Factor 1 alpha (HIF-1α) signaling, and glycolysis. In situ hybridization (ISH) and immunohistochemical (IHC) analyses demonstrated that many of the deregulated genes, including the HIF-1α targets *Vegfa*, *Slc2a1* (encodes glucose transporter 1, Glut1), and *Ldha* (encodes Lactate dehydrogenase A), are expressed by early RG. We further observed that *Vegfa* transcript and protein are endogenously upregulated in RG as Foxp1 levels decline, and these changes coincide with the onset of angiogenesis. Moreover, the deletion of Foxp1 from early cortical progenitors resulted in early destabilization of HIF-1α protein, upregulation of HIF-1α targets, including Glut1, Ldha, and Vegfa, and accelerated development of the cortical vasculature. Finally, our in vitro studies demonstrate that HIF-1α stabilization is sufficient to rescue the premature differentiation of Foxp1-deficient neural progenitor cells (NPCs). These findings reveal that Foxp1 attenuates the HIF-1α signaling pathway, suppressing angiogenesis and prolonging conditions that sustain early RG character.

## Results and discussion

### Early loss of Foxp1 in radial glia upregulates genes associated with neuronal differentiation, angiogenesis, HIF-1 signaling, and glycolysis

To examine the role of Foxp1 directing the transition from self-renewal to neurogenic divisions, we dissected samples of the lateral cortex (within the presumptive somatosensory cortex) from embryonic day (E)12.5 control (*Emx1^Cre* negative littermates) and *Foxp1^fl/fl*; *Emx1^Cre/+* (termed *Foxp1^cKO*) mutant embryos and performed bulk RNA-Seq analysis (Figs. 1A and EV1A–C). We previously demonstrated that Cre recombination occurs at E10.5, and significant Foxp1 protein loss occurs at E11.5 (Pearson et al, 2020). Of the 514 significantly misregulated genes, 307 were upregulated ($\log_2$ fold change >0.5), and 80 were downregulated ($\log_2$ fold change < −0.5), consistent with the reported role of Foxp1 as a transcriptional repressor in other tissues (Zhang et al, 2010) (Fig. EV1A and Tables EV1–EV2). The main human disease categories associated with these upregulated genes included schizophrenia, autism, intellectual disorders, neurodevelopmental disorders, glycogen storage disorders, and seizures (Fig. EV1B). Gene ontology analyses showed that terms associated with DNA replication, cell cycle/mitosis, and RNA metabolism were underrepresented in the absence of Foxp1 (Fig. 1B). Concordantly, subcellular compartments associated with the top downregulated genes included the cyclin E1-CDK2 complex, the replication fork, and the mitochondrion (Table EV3). Without Foxp1, biological processes such as nervous system development, neuron differentiation, and cell–cell signaling were overrepresented (Fig. 1B). In addition, among the most overrepresented processes and pathways in Foxp1^cKO cortices were responses to cell-cell signaling, regulation of blood circulation, glycolysis/gluconeogenesis, and HIF-1 signaling (Fig. 1B). Subcellular compartments associated with the upregulated genes include the 6-phosphofructokinase complex, glucose transporter complex, and synapses (Table EV4).

Consistent with these findings, we found that many essential genes involved in glycolysis, HIF-1 signaling, and angiogenesis were upregulated in the absence of Foxp1, including *Vegfa*, *Ldha*, and *Slc2a1* (Fig. 1C). Thus, early loss of Foxp1 in RG leads to

transcriptional increases in genes associated with differentiation, angiogenesis, and increased dependence on metabolites such as oxygen and glucose.

### The endogenous downregulation of Foxp1 coincides with the elaboration of the cortical vasculature

Given the increased expression of genes connected to processes such as blood circulation in Foxp1 mutants, we asked whether Foxp1 downregulation during normal cortical development coincided with changes in cortical vasculature. IHC analysis of Foxp1 protein levels in RG and the presence of Isolectin IB4$^+$ blood vessels, followed by surface rendering of the blood vessels, enabled us to distinguish periventricular plexus (PVP) vessels from those originating from the perineural vascular plexus (PNVP). Our analyses confirmed that there are few PVP vessels at E12.5 when Foxp1 levels are highest (Fig. 1D,G,H). As previously reported (Vasudevan et al, 2008), the cortex is perfused ventrodorsally (Fig. 1D). Foxp1 downregulation at E13.5 coincides with the onset of angiogenesis in the cortex, i.e., an increased number of vessels per section, followed by an increase in PVP vessel volume at E14.5 (Fig. 1E–I). At these time points, we observed a significant increase in Tbr2$^+$ IPs and cortical plate thickness as TUJ1$^+$ neurons were generated (Figs. 1E,F,J,K and EV1D–F).

### Foxp1 attenuates the HIF-1α signaling response in radial glia

Previous reports demonstrated that HIF-1α expression is destabilized between E11.5 and E12.5 (Lange et al, 2016). Using IHC, we analyzed HIF-1α expression and found it is expressed in Nestin$^+$ RG at E11.5 when Foxp1 expression is low (Fig. 2A–D). By the next day (E12.5), HIF-1α is downregulated in the cortex as Foxp1 expression increases (Fig. 2E–H). HIF-1α expression was detected in Nestin$^+$ RG at low levels in the dorsal-most lateral cortex and higher levels in the medial cortex (Fig. 2I,J). This pattern is consistent with the ventral to dorsal vascularization of the cortex. In Foxp1^cKO cortices at E12.5, while HIF-1α expression in the dorsolateral cortex remained low/absent, there was a significant decrease in HIF-1α staining in the medial cortex (Fig. 2K–N).

Next, we sought to determine whether RG express key HIF-1α targets at E12.5. From our RNA-Seq dataset, we selected HIF-1α targets involved in glycolysis and analyzed expression by ISH. Several glycolysis genes are specifically expressed in the VZ, including *Slc2a1*, *Ldha*, *Aldoa*, *Pfkl*, *Pfkfb3*, *Pdk1*, and *Slc16a4* (Fig. EV2A–G). Western blot analyses of E11.5, E12.5, and E13.5 wild-type cortex lysates demonstrate HIF-1α protein levels decrease while Glut1 levels increase (Fig. EV2H–J). Ldha levels decrease between E11.5 and E12.5 but remain stable between E12.5 and E13.5 (Fig. EV2H,K). Vegfa levels similarly decrease between E11.5 and E12.5, though slightly increase between E12.5 and E13.5 (Fig. EV2H,L). A caveat to this analysis is that protein lysates include RG and other cell types in the cortex at each stage, including endothelial cells. Thus, we performed IHC with each antibody and Nestin to better assess protein abundance in RG.

IHC analysis showed that Glut1 protein is expressed in Nestin$^+$ RG with pronounced accumulation at their apical end feet at E12.5 and E13.5 (Fig. EV2M,O,Q–T). Additional Nestin co-staining analysis confirmed Ldha expression in RG (Fig. EV2N,P,U–X). Thus, several HIF-1α targets, including Glut1 and Ldha, persist in RG, while HIF-1α expression itself is downregulated between E11.5 and E12.5.

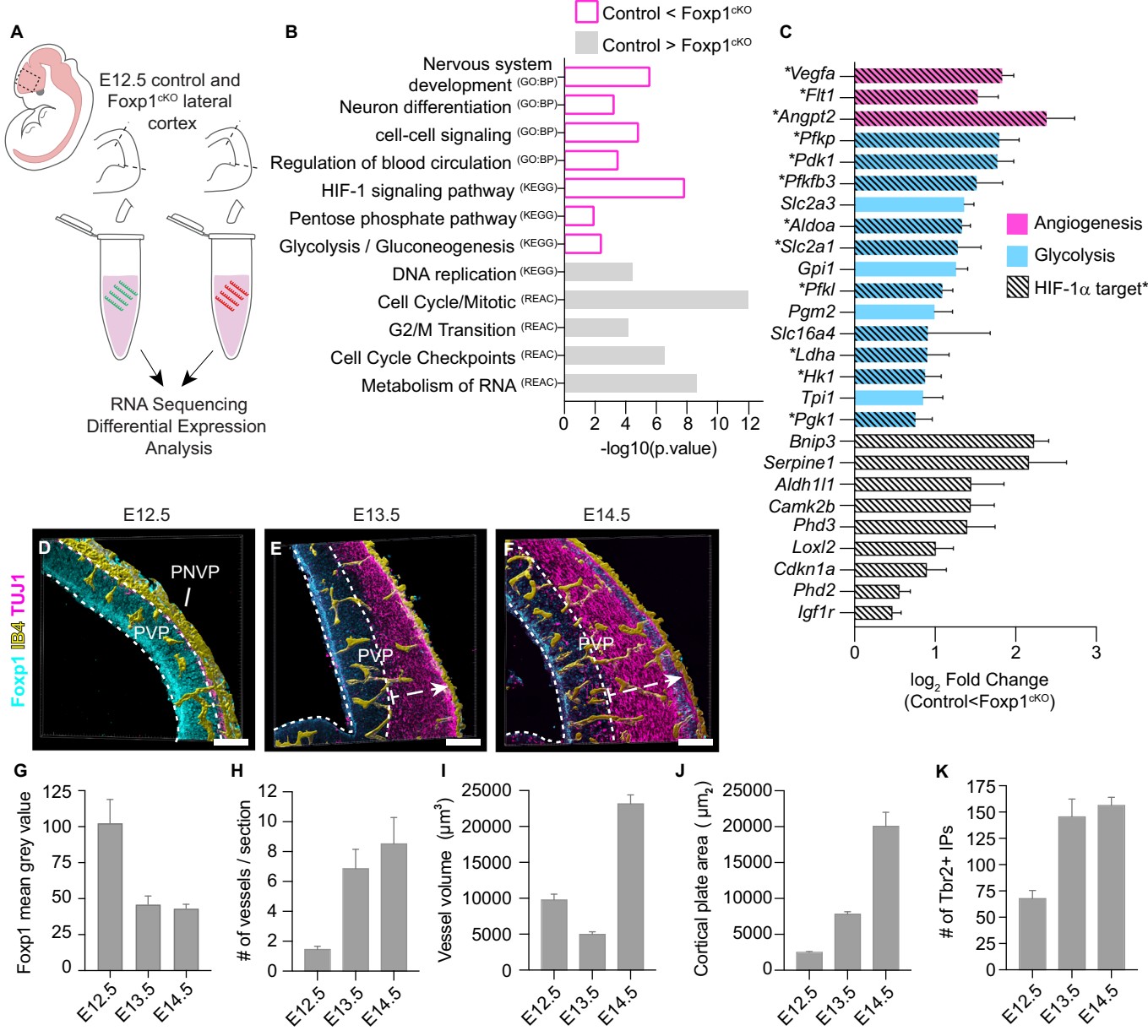

**Figure 1. Processes regulating angiogenesis, HIF-1 signaling, and glycolysis are upregulated upon conditional removal of Foxp1.**

(A) RNA collected from E12.5 control and Foxp1cKO cortices. (B) Gene ontology terms associated with significantly misregulated genes in the Foxp1cKO cortex at E12.5. GO terms: BP, biological process; KEGG, pathways, REAC, reactome pathway. (C) Significantly misregulated genes associated with angiogenesis (magenta), glycolysis (blue), and HIF-1α signaling (white/hashed lines) in Foxp1cKOs at E12.5. * and bars with hashed lines are also HIF-1α targets. (D–F) 3D surface rendering of IB4+ blood vessels in the cortex from E12.5 to E14.5 with Foxp1 and TUJ1 expression. White dashed arrows delineate cortical plate. (G–J) Foxp1 intensity, periventricular plexus vessel (PVP) number, volume, and cortical plate area in the lateral cortex at E12.5–E14.5. (K) Number of Tbr2+ progenitors (per 200 μm²) in wild-type cortex at E12.5–E14.5. Scale bars 100 μm. Data information: Statistical significance determined by ANOVA (B, C). N = 4 controls, 2 mutants (B, C). N = 5 (G), 4 (H), 3–4 (I), 3–4 (J), 3–4 (K) embryos/time point. All data represented as mean ± SEM. Source data are available online for this figure.

To explore the connection between Foxp1 and HIF-1α signaling, we next confirmed by qPCR that several key HIF-1α targets are upregulated at E12.5 in the absence of Foxp1, including *Slc2a1, Ldha, Aldoa, Pfkl, Pfkfb3, Pdk1,* and *Slc16a4* (Fig. 2O). We subsequently analyzed Glut1 and Ldha protein levels and found that both proteins were significantly increased in Nestin+ RG lacking Foxp1 function (Fig. 2P–S,V,W). Thus, while HIF-1α targets are elevated in the absence of Foxp1, HIF-1α protein

becomes reduced. These data suggest that Foxp1 typically acts to stabilize HIF-1α protein yet attenuate its signaling functions. These opposing actions of Foxp1 on HIF-1α are not without precedent, as previous studies have demonstrated that prolonged hypoxia can trigger negative feedback loops where HIF-1α protein becomes unstable or is expressed in a pulsatile manner (Nguyen et al, 2013; Bagnall et al, 2014). Likewise, some studies have shown that the Cdk2/cyclin E cell cycle regulators can inhibit HIF-1α protein levels

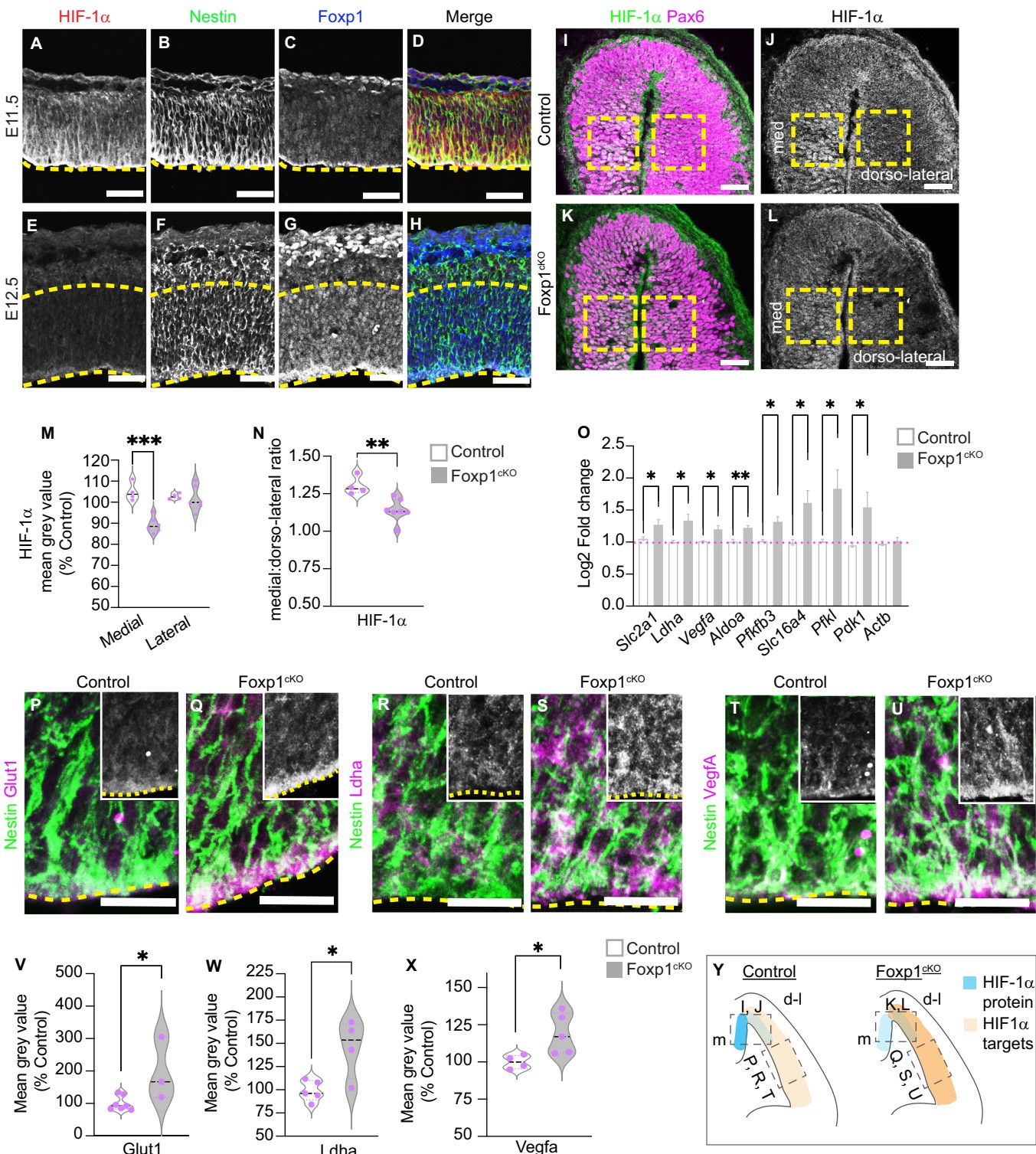

## Vegfa upregulation in Foxp1-deficient radial glia

while increasing HIF1 transcriptional activity (Sengupta et al, 2011; Hubbi et al, 2014). This latter observation may be relevant as our RNA-seq analysis singled out the Cdk2/cyclin E complex as among the most changed cellular compartments in Foxp1 mutant cortices (Tables EV2 and 3).

The HIF-1α target gene Vegfa is a critical mediator of cortical angiogenesis and among the most upregulated genes in our RNA-seq analysis (Figs. 1C and EV1A). RNAScope analysis of Vegfa and

**Figure 2. HIF-1α expression is reduced, and HIF-1α targets are upregulated in RG in the absence of Foxp1.**

(A–H) IHC analysis of HIF-1α and Foxp1 in Nestin⁺ RG at E11.5 and E12.5 wild-type cortex. (I–L) IHC analysis of Pax6 and HIF-1α expression in the medial and dorsolateral cortex at E12.5 in control and Foxp1cKO mutants. Boxed areas indicate regions quantified in (M) and (N). (M, N) HIF-1α mean gray value (percent control) and medial to dorsal ratio in control and Foxp1cKO mutant cortex at E12.5. (O) Quantification of mRNA fold enrichment (normalized to *Actb*) for *Slc2a1, Ldha, Vegfa, Aldoa, Pfkfb3, Slc16a4, Pfkl, Pdk1,* and *Actb* in control and Foxp1cKO by qPCR in the lateral cortex at E12.5. (P–U) IHC for Glut1, Ldha, Vegfa, and Nestin in control and Foxp1cKO mutants in the VZ of the cortex at E12.5. The inset is Glut1/Ldha/Vegfa only. (V) Glut1 mean gray value (percent control) in control and Foxp1cKO mutant Nestin⁺ RG at E12.5. (W) Ldha mean gray value (percent control) in control and Foxp1cKO cortex at E12.5. (X) Vegfa mean gray value (percent control) in control and Foxp1cOn cortex at E12.5. (Y) Summary of HIF-1α and target gene expression in control and Foxp1cKO cortex at E12.5. Dashed lines demarcate the ventricular zone. Scale bars 50 μm (A–L), 20 μm (P–U). Data information: p = 0.0003 and 0.0055, respectively, Student's t-test. N = 7 control, 6 mutants (M, N). N = 3 control, 3 mutants (3 litters). p values = 0.039 (*Slc2a1*), 0.0256 (*Ldha*), 0.013 (*Vegfa*), 0.0073 (*Aldoa*), 0.0177 (*Pfkfb3*), 0.0152 (*Slc16a4*), 0.0183 (*Pfkl*), 0.0392 (*Pdk1*). Student's t-test (O). p = 0.0262, Student's t-test. N = 7 control, 3 mutants (V). p = 0.0169, Student's t-test. N = 5 control, 4 mutants (W). p = 0.0328, Students t-test. N = 4 control, 5 mutants (X). All data represented as mean ± SEM. Source data are available online for this figure.

*Foxp1* expression demonstrated the upregulation of *Vegfa* expression in the VZ at E13.5 and E14.5, whereas *Foxp1* is downregulated. *Vegfa* and *Foxp1* are also both expressed in the cortical plate (Fig. EV3A–D). IHC analysis of Vegfa protein in wild-type tissue between E12.5 and E14.5 confirmed its upregulation in Nestin⁺ RG from E13.5 onwards, with accumulation at the apical surface of the VZ (Fig. EV3E–K). Co-staining analysis of Vegfa protein with Nestin and Foxp1 further showed that E12.5 Nestin⁺ RG express high levels of Foxp1 and low levels of Vegfa (Fig. EV3L–O) while E14.5 Nestin⁺ RG express low levels of Foxp1 and increased levels of Vegfa (Fig. EV3P–S).

To determine whether Vegfa levels were affected by the loss of Foxp1 in RG, we compared the intensity of Vegfa expression in Nestin⁺ RG in control and Foxp1cKO cortex at E12.5 by IHC. This analysis confirmed that Vegfa is increased in Foxp1cKO RG (Fig. 2T,U,X). The transcriptional increase in Vegfa expression was demonstrated by qPCR analysis in the control and Foxp1cKO cortex at E12.5 (Fig. 2O). IHC analysis of Vegfa expression levels in the cortical plate showed no significant difference between control and Foxp1cKO cortices (Fig. EV3T–V). Thus, the upregulation of Vegfa within RG appears to coincide with the endogenous downregulation of Foxp1 and increased angiogenesis in the cortex. Premature loss of Foxp1 leads to an increase in Vegfa expression in RG (Fig. 2Y).

## Deletion of Foxp1 accelerates the development of the cortical vasculature

Given the upregulation of Vegfa and the downregulation of HIF-1α in the absence of Foxp1, we asked whether the loss of Foxp1 impacted cortical angiogenesis. Initially, we examined how this manipulation influenced the ventrodorsal expansion of the PVP network within the E12.5 cortex using surface-rendered projections of IB4⁺ vessels in 50-μm-thick cryosections of control and Foxp1cKO embryos. At E12.5, comparable numbers of vessels were present in the mid and ventrolateral cortex but appeared to increase in the dorsolateral region of Foxp1cKO samples (Fig. 3A–D). To quantify these differences, we divided the cortex into three areas of equal size. We counted the number of vessels in each region, which confirmed a significant increase in vessels in the dorsolateral area (Fig. 3E,F). Thus, increased Vegfa expression and HIF-1α down-regulation coincided with increased vasculature in the Foxp1-deficient dorsolateral cortex. While IB4 can be expressed by other cell types, including microglia in the developing brain (Cunningham et al, 2013; Penna et al, 2021), at the stages we investigated, we found that IB4 exclusively labeled CD31⁺ endothelial cells

(Appendix Fig. S1A–F). However, IB4-labeling of filopodia was more evident than CD31 and easier to visualize and quantify (Appendix Fig. S1G,H). We observed a higher density of tip cell filopodia in Foxp1cKO samples at E12.5 compared to controls (Fig. 3G,H,M), which is consistent with previous observations that increased Vegfa signaling can induce the formation of endothelial tip cell filopodia (Gerhardt et al, 2003; Haigh et al, 2003).

We next analyzed the cortical vasculature a day later at E13.5. At this stage in control embryos, the cortex has been perfused by contiguous IB4⁺ vessels along the dorsoventral axis that have begun to form branches (Fig. 3I,K). Parallel analysis of Foxp1cKO cortices showed that their blood vessels were significantly more extended and branched, and volume increased compared to controls (Fig. 3J,L,N–P). Collectively, these results demonstrate that deletion of Foxp1 results in a premature establishment of cortical vasculature in the dorsal-most lateral cortex and the development of the vascular network in a non-stereotypical fashion.

Other signals have been shown to regulate cortical angiogenesis, including lactate and Wnts (Daneman et al, 2009; Dong et al, 2022). Wnt genes were not significantly misregulated in our RNA-Seq analyses, suggesting that Foxp1 may act independently of Wnt signaling to influence angiogenesis. Foxp1 is also expressed in neurons, as is Vegfa (Fig. EV3A–D), raising the possibility that at later stages, loss of Foxp1 in neurons may also influence angiogenesis. The changes we have shown here in Ldha and other glycolysis genes in the absence of Foxp1 could reflect the increased availability of glucose in the RG microenvironment. Mouse models of maternal diabetes have shown that moderate hyperglycemia leads to premature RG differentiation (Ji et al, 2019). Thus, increases in environmental glucose could also influence the balance between RG self-renewal and differentiation (Andrews and Pearson, 2024).

## HIF-1α stabilization rescues early differentiation phenotypes in Foxp1-deficient NPCs

Given our results, we hypothesized that Foxp1 promotes RG maintenance by stabilizing HIF-1α and attenuating the HIF-1α signaling response to promote hypoxic conditions that favor self-renewal. To test this hypothesis, we generated cortical spheroids from control and Foxp1KO mouse embryonic stem cells (mESCs). This in vitro approach also enabled us to determine the cell-autonomous effects of Foxp1. We used two Foxp1KO mESC lines in which the genomic region encoding the Foxp1 forkhead domain required for DNA binding was deleted using CRISPR-Cas9 editing (Appendix Fig. S2). This modification caused a reduction in Foxp1 protein levels and truncated the sizes of both isoforms A and D

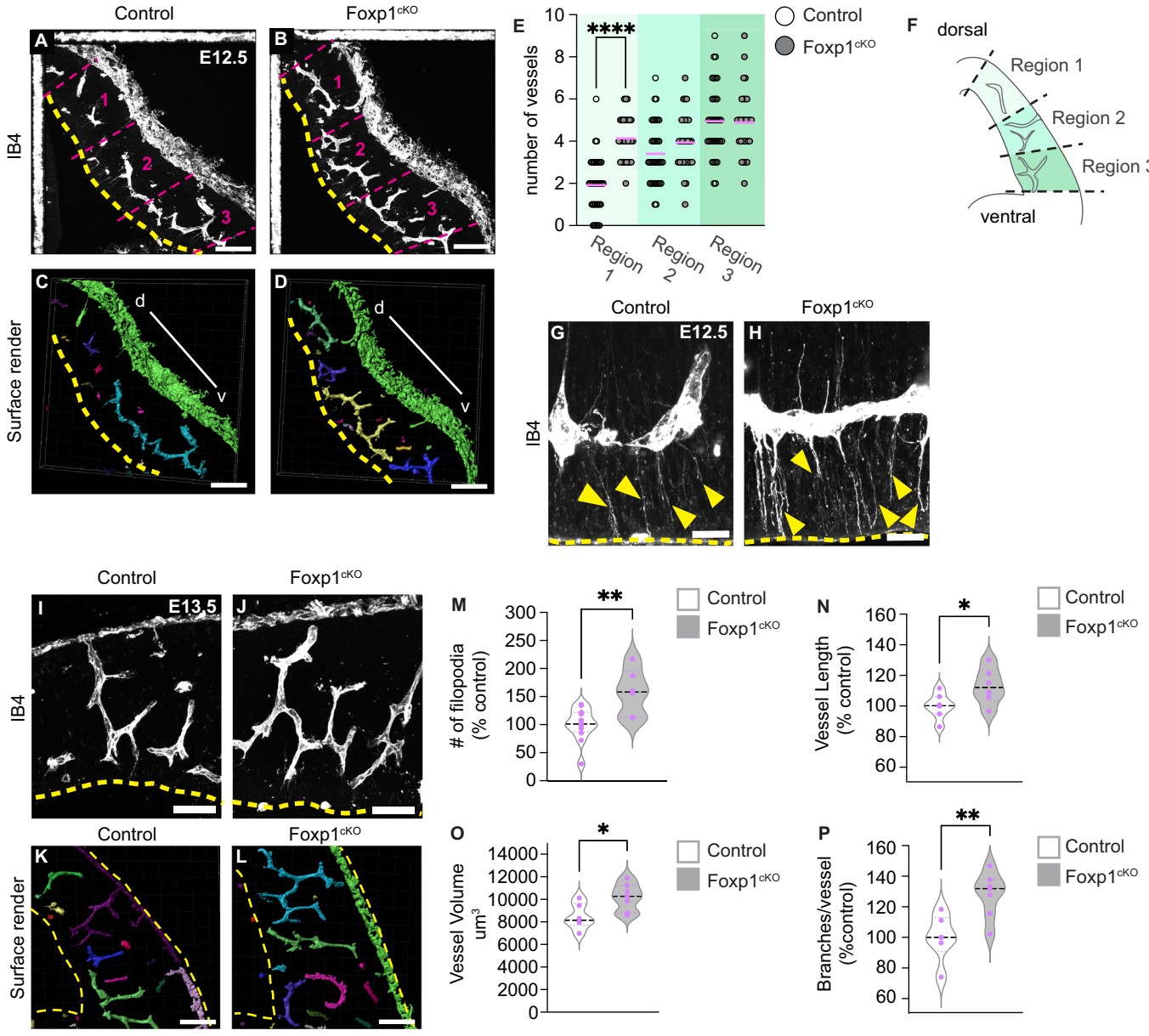

**Figure 3. Loss of Foxp1 results in precocious development of the cortical vasculature.**

(A, B) Isolectin B4 (IB4) vessel staining in the cortex of control and Foxp1cKO embryos at E12.5. (C, D) Surfaced rendered contiguous vessels in (A) and (B). (E) Number of IB4+ vessels in each binned area in control and Foxp1cKO lateral cortices at E12.5. (F) Schematic of bins used for quantification in (E). (G, H) IB4 labeled filopodia at the ventricular surface in control and Foxp1cKO cortex at E12.5. Arrowheads mark filopodia. (I–J) IB4+ vessels in control and Foxp1cKO cortex at E13.5. (K, L) Surface rendered cortex images in control and Foxp1cKO at E13.5. (M) Mean filopodia (percent control) in control and Foxp1cKO cortex at E12.5. (N–P) Vessel length, vessel volume, and number of branches per vessel in control and Foxp1cKO cortex. Dashed yellow lines demarcate the apical surface. d, dorsal; v, ventral. Scale bars 100 µm (A–D), 5 µm (G, H), and 50 µm (I–L). Data information: $p < 0.0001$, Mann–Whitney test. $N = 10$ control, 7 mutants (E). $p = 0.0033$, Student's t-test. $N = 12$ control, 6 mutants (M). $p = 0.0386$, 0.0235, and 0.0063, respectively, Student's t-test. $N = 6$ control, 7 mutants (N–P). All data represented as mean ± SEM. Source data are available online for this figure.

(Fig. 4A). Spheroids at day 7 in vitro (div) were mainly composed of Pax6+ NPCs, and very few cells expressed neuronal markers such as Tbr1 or Ctip2, with little difference seen between control and Foxp1KO samples (Fig. EV4A–F). The loss of Foxp1 protein in Pax6+ NPCs was confirmed by IHC (Fig. EV4H–P). By 10 div, we observed Foxp1 loss in spheroids recapitulates the in vivo phenotype. Western blot analyses showed that HIF-1α protein levels are decreased in the absence of Foxp1 (Fig. 4B). qPCR

analysis demonstrated that the expression of HIF-1α target genes is increased in Foxp1KO spheroids, except for *Pfkfb3* which was decreased in Foxp1KO#1 (Figs. 4C and EV4G). In addition, we observed a significant decrease in the number of Pax6+ NPCs and a concomitant increase in Ctip2+ neurons in spheroids generated from both Foxp1KO lines (Fig. 4E–L).

To test whether the precocious differentiation phenotype seen in the Foxp1KO spheroids could be rescued by HIF-1α manipulation,

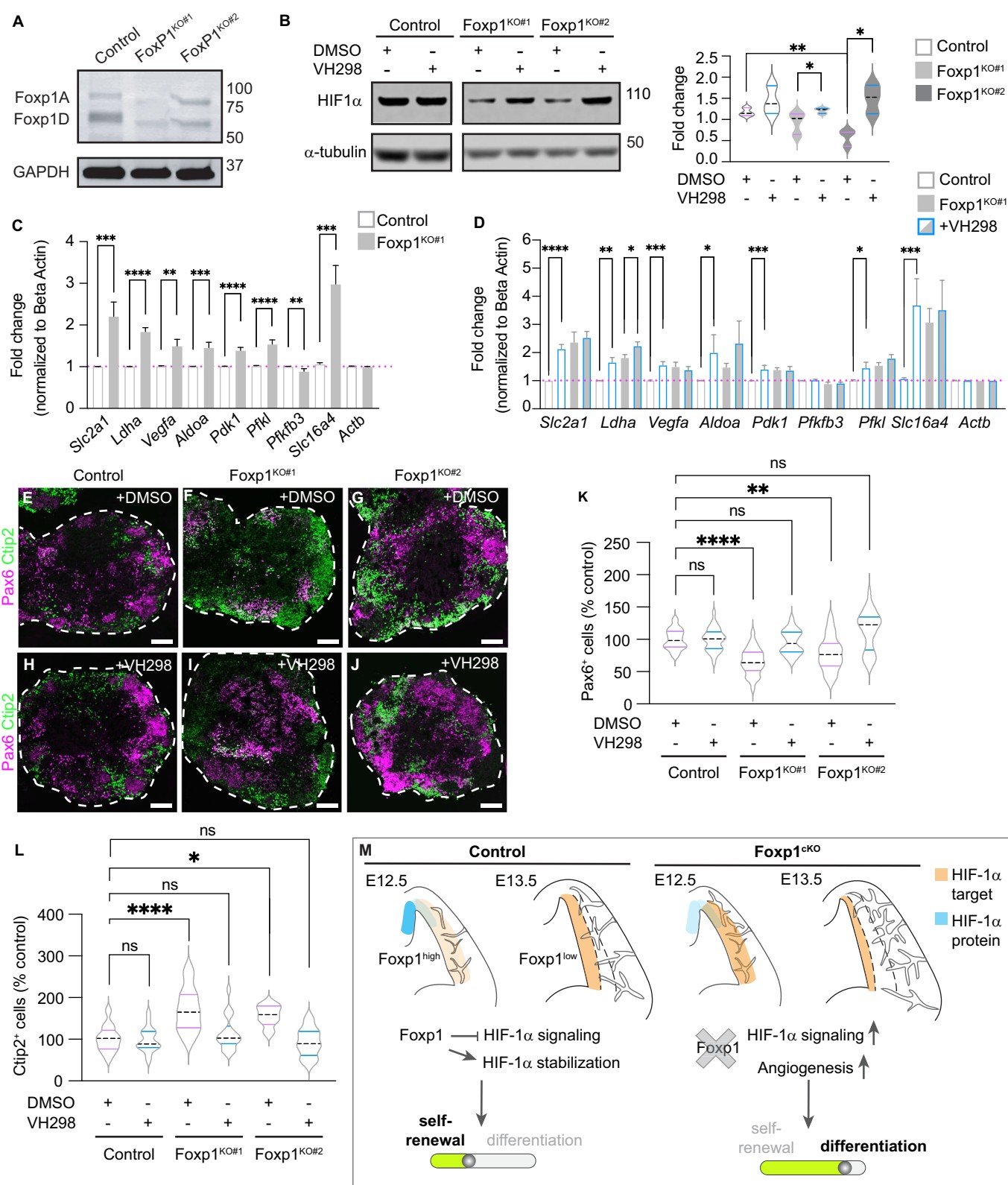

we treated spheroids at 7 div with VH298, a Von Hippel-Lindau E3 ligase inhibitor that stabilizes HIF-1α (Frost et al, 2016). Western blot analyses confirmed the stabilization of HIF-1α in control and Foxp1^KO spheroids treated with VH298, and the treatment

correlated with increased expression of HIF-1α target genes in control spheroids (Fig. 4B,C). In Foxp1^KO spheroids, the transcriptional response to VH298 was markedly reduced or showed no change compared to the response in control spheroids, except for

**Figure 4. HIF-1α stabilization promotes NPC maintenance in Foxp1-deficient cortical spheroids.**

(A) Western blot analysis of Foxp1 and GAPDH protein in control and Foxp1KO mouse embryonic stem cells. (B) Western blot analysis and quantification of HIF-1α protein (compared to α-tubulin) in control and Foxp1KO spheroids at 10 div treated with DMSO or VH298. (C) qPCR analysis of HIF-1α target gene expression in control and Foxp1KO#1 spheroids at 10 div. (D) qPCR analysis of HIF-1α target gene expression in control and Foxp1KO#1 spheroids at 10 div. treated with DMSO or VH298. (E–J) IHC for Pax6+ NPCs and Ctip2+ neurons in control and Foxp1KO spheroids (10 div) treated with DMSO or VH298. (K) Percentage of DAPI+ cells that are Pax6+ in control and Foxp1KO spheroids (10 div) treated with DMSO or VH298. (L) Percentage of DAPI+ cells that are Ctip2+ in control and Foxp1KO spheroids (10 div) treated with DMSO or VH298. (M) Schematic of control and Foxp1cKO phenotypes. Foxp1 attenuates the HIF-1α signaling pathway to promote RG self-renewal. In the absence of Foxp1, HIF-1α target gene expression is upregulated, angiogenesis is perturbed, and RG precociously differentiate. Scale bars 100 μm. Data information: $p =$ **0.0097, 0.00364 (Foxp1KO#1) and 0.0162 (Foxp1KO#2). Student's t-test. $N = 10$–12 spheroids, western blots run in triplicate (B). $p$ values = 0.0008 (Slc2a1), <0.0001 (Ldha), 0.0026 (Vegfa), 0.0010 (Aldoa), <0.0001 (Pdk1), <0.001 (Pfkl), 0.0061 (Pfkfb3, Mann–Whitney test), 0.0001 (Slc16a4). $N = 10$–12 spheroids/experiment, 3 experiments (C). $p$ values = <0.0001 (Slc2a1, control), 0.0013 (Ldha, control), 0.0446 (Ldha, Foxp1KO#1), 0.0002 (Vegfa, control), 0.0179 (Aldoa, control), 0.0002 (Pdk1, control), 0.0201 (Pkfl, control), 0.0003 (Slc16a4, control). $N = 10$–12 spheroids/experiment, 3 experiments (D). One-way ANOVA, $p =$ >0.9999, <0.0001, 0.928, 0.0024 and 0.1066, respectively. $N = 10$–12 spheroids/experiment, 3 experiments (K). Ordinary one-way ANOVA, $p =$ >0.9999, <0.0001, 0.8306, 0.0196, 0.9883. $N = 10$–12 spheroids/experiment, 3 experiments (L). All data represented as mean ± SEM. Source data are available online for this figure.

Ldha in Foxp1KO#1 and Pdk1 in Foxp1KO#2 spheroids, which exhibited significant increases (Figs. 4D and EV4Q). These effects may reflect the reduced HIF-1α levels seen in Foxp1KO spheroids compared to controls. Additional analysis of the percentage of cells that are Pax6+ NPCs or Ctip2+ neurons demonstrated a significant increase in NPCs in Foxp1KO spheroids treated with VH298 and a concomitant decrease in Ctip2+ neurons compared to DMSO-treated Foxp1KO spheroids (Fig. 4E–L). Moreover, after VH298 treatment, the proportion of Pax6+ NPCs and Ctip2+ neurons in Foxp1KO spheroids was similar to DMSO-treated control spheroids. Thus, HIF-1α stabilization appears sufficient to rescue defects in NPC maintenance and premature differentiation associated with Foxp1 deficiency.

Integrating these findings with our prior investigations into the effects of Foxp1 gain- and loss-of-function on cortical progenitor activities (Pearson et al, 2020), our studies suggest that Foxp1 shapes the transition of RG through early-to-late stages of neurogenesis by promoting a hypoxic stem cell niche microenvironment. Early on, Foxp1 suppresses the downstream effectors of HIF-1α, such as Vegfa, that lead to relief from hypoxia, thus delaying angiogenesis and the transition to oxygenated conditions. This suppression promotes an environment favorable for early self-renewing cell divisions. As Foxp1 levels in RG endogenously decline, or when Foxp1 function is experimentally ablated, HIF-1α target gene expression levels increase, and HIF-1α is further destabilized. These effects drive the elaboration of the cortical vasculature and the switch to neurogenic divisions (Fig. 4M). These findings complement and extend previous reports demonstrating that hypoxic conditions and the suppression of vascular ingrowth repress the switch to neurogenesis (Lange et al, 2016; Komabayashi-Suzuki et al, 2019; Dong et al, 2022).

The cellular response to hypoxia activates a transcriptional cascade that drives processes that enable a transition to a normoxic environment and processes that promote cell survival and proliferation (Corrado and Fontana, 2020). Factors have been identified that selectively regulate cellular responses to hypoxia, including Runt-related (Runx) transcription factors that modulate HIF-1α stability and transcriptional activity to regulate Vegfa expression during angiogenesis (Peng et al, 2008; Lee et al, 2012, 2014; Brocato et al, 2014). Therefore, cells can dissociate the proliferative effects of HIF-1α signaling and the adaptive effects, such as glycolysis and angiogenesis. Our in vitro experiments reveal an intrinsic role for Foxp1 in promoting cell proliferation and repressing angiogenesis and glycolysis, identifying Foxp1 as a key regulator of the HIF-1α signaling pathway in early RG.

Presently, the mechanisms driving Foxp1 downregulation in RG cells across development are unknown, though several factors could be involved. HIF-1α has been shown to directly bind and regulate the expression of Jumonji domain-containing histone demethylases demethylases such as Jarid1b (Kdm5b), and studies have shown that the range of HIF-1α targets is determined by cell type specific patterns of chromatin structure (Brocato et al, 2014). Recently, a related factor, Jarid2, was identified as a repressor of early Foxp1 in retinal progenitors (Zhang et al, 2023). Further studies are required to determine whether a similar mechanism is utilized in RG. Another regulatory mechanism may involve microRNAs, which have been shown to regulate Foxp proteins in many organ systems (Li et al, 2017). miR-9 has been demonstrated to inhibit factors that regulate Vegfa expression in retinal and telencephalic neurons, coupling neurogenesis and the maturation of the vasculature (Madelaine et al, 2017). Foxp1 is a target of miR-9 in a variety of contexts, including in motor neurons in the developing spinal cord (Otaegi et al, 2011; Gomez et al, 2014; Radhakrishnan and Alwin Prem Anand, 2016; Jiang et al, 2017). Potentially, a similar miR-9-dependent mechanism could be used to regulate Foxp1 and Vegfa expression in RG.

An increasing body of evidence has implicated blood vessel pathologies in several neurodevelopmental disorders (Baruah and Vasudevan, 2019; Ouellette and Lacoste, 2021). For example, early defects in PVP endothelial cells have been linked to the origin of autism (Azmitia et al, 2016). Further investigation is required to determine whether the mild alterations we observe in the Foxp1cKO cortex can have long-term effects on brain function. Foxp1 is also expressed in endothelial cells and has been shown to promote angiogenesis in various contexts (Grundmann et al, 2013). Further dissection of the neural and endothelial contributions of Foxp1 to brain angiogenesis will be critical for understanding the role of FOXP1 in neurodevelopmental disorders.

## Methods

### Mouse lines

C57BL6 wild-type (RRID: IMSR_JAX:000664), Foxp1flox/flox (RRID: IMSR_JAX:017699), and Emx1Cre (RRID: IMSR_RBRC01342) mice were maintained as previously described (Iwasato et al, 2004; Feng et al, 2010) following UCLA Chancellor's Animal Research Committee husbandry guidelines. All experiments were conducted

in accordance with relevant NIH guidelines and regulations, related to the Care and Use of Laboratory Animals tissue. Animal procedures were performed according to protocols approved by the Research Animal Resource Center at Weill Cornell College of Medicine. Wild-type litters were provided by the Mouse Genetics Core Facility at Memorial Sloan Kettering Cancer Center. Male and female embryos between embryonic days 11.5 and 13.5 were used in this study. Cre-negative littermates were used as controls.

## Foxp1 knock out mouse embryonic stem cell line generation

The Foxp1 gene was disrupted in MM13 mouse embryonic stem cells (mESCs). To disrupt Foxp1 function, CRISPR/Cas9-mediated genome editing was used to target the forkhead domain. Using Benchling, two guides were designed targeting Exon 12 of Foxp1-201 (ENSMUST00000074346): 5′-AGATTCGAGAATGGCCTACG-3′ and 5′-TGCAAAGCTTACCTTCCACG-3′. These sequences were individually cloned into plasmid pSpCas9(BB)-2A-Puro (PX459) V2.0 (Addgene #62988) and then electroporated into the mESCs using the Lonza Nucleofector 2b Device and Cell Nucleofector Kit (Lonza #VAPH-1001). The mESCs were cultured in feeder-free 2i media (DMEM/F-12 (Gibco #11320033) and Neurobasal Medium (Gibco #21103049) (prepared at a 1:1 ratio) supplemented with N2 supplement (Gibco #17502048), B27 serum-free supplement (Gibco #17504044), penicillin/streptomycin (Gibco #15140122), glutaMAX (Gibco #35050061), bovine serum albumin (Thermo Scientific Chemicals #AAJ64248AE), 55 μM 2-mercaptoethanol (Gibco #21985023), 3 μM CHIR99021 (Axon #1386), 1 μM PD 98059 (Tocris Bioscience #1213), and ESGRO recombinant mouse LIF protein (1000 units/ml, MilliporeSigma #ESG1107)]. Antibiotic selection was initiated 24 h after nucleofection in 2i media containing 1.5 μg/ml puromycin (Fisher BioReagents #BP2956100) for ~72 h. Clones were picked individually, expanded in feeder-free 2i media, and genomic DNA was collected using QuickExtract DNA extraction solution (Lucigen, #QE09050). The region surrounding the targeted site was PCR-amplified (using the primers Forward 5′-TTTGTGAAACCTGCCTGAGGA-3′ and Reverse 5′-GGTCACAAGGTCACCTCCTT-3′) to visualize a successful deletion facilitated by the guides. A loss of full-length FOXP1 protein was then verified by Western blot using a protein-specific antibody (Cell Signaling Technology, #4402T).

## Mouse embryonic stem cell maintenance

MM13 mESCs were maintained in 2i mESC media on 0.1% gelatin-coated plates. DMEM/F12 (50%) and Neurobasal (50%) basal media was supplemented with 0.5% N2, 1% B27, 1% penicillin-streptomycin, 1% Glutamax, beta mercaptoethanol. Cells were fed daily, media was supplemented with 10 ng/ml human LIF, 10 μM CHIR, 1 μM PD98059. mESCs were routinely tested for mycoplasma.

## Cortical neural progenitor cell and neuronal differentiation

Cortical neural progenitor cells and neurons were generated using the protocol established by Eiraku et al (Eiraku and Sasai, 2011). In brief, 3000 mESCs were plated per well of a 96-well U-bottomed ultra-low attachment plates. Cells were cultured in cortical differentiation media (GMEM, non-essential amino acids, pyruvate, beta-mercaptoethanol, knockout replacement serum) for

7 days. Half-media changes were performed daily. On day 7 spheroids were transferred to cortical maturation media (DMEM/F12, Glutamax, N2, penicillin-streptomycin) in a plastic culture dish for 3 days.

## Tissue preparation

Embryonic cortices were dissected and fixed in 4% paraformaldehyde in PBS overnight. Tissues were cryosectioned (10–50 μm sections) and processed for immunohistochemistry, in situ hybridization, or RNAScope fluorescent in situ hybridization. For immunohistochemistry, spheroids were fixed in 4% paraformaldehyde for 30 min on ice. After three washes in PBS, Spheroids were transferred to 30% sucrose for approximately 2 h on ice (until spheroids sink). Spheroids were cryosectioned at 8–10 μm.

## RNA sample collection

E12.5 lateral cortex samples were lysed in QIAzol reagent, and RNA was extracted following the manufacturer's instructions (miRNeasy Micro Kit, Qiagen). Six samples were used for RNA sequencing: four control females and two Foxp1[cKO] females. RNA concentration and integrity were assessed with Agilent RNA ScreenTape analysis using the Agilent 2100 Bioanalyzer. All samples used in downstream analyses had a RIN > 9.7. Spheroids were lysed in Qiazol reagent and RNA was extracted following manufacturer's instructions (RNAeasy Micro Kit, Qiagen).

## RNA sequencing and analysis

RNA samples were sent to the UCLA Neuroscience Genomics Core for library preparation and sequencing. Sample libraries were generated using TruSeq Stranded RNA kit and sequenced with paired-end 150 base pair reads on two lanes using the Illumina HiSeq 3000. Each sample contained 58–91 million reads (an average of 79 million). All samples of raw sequence data passed quality control using FastQC (Andrews, 2010). The data was mapped to the mouse MM10 genome (Gencode version 17) using STAR aligner (Dobin et al, 2013) with default parameters. MultiQC was used to aggregate quality metrics produced by STAR (Ewels et al, 2016). Within each sample, 92–93% of reads were uniquely mapped and used for further analyses. BAM files produced by STAR were sorted and converted to SAM files using samtools (Li et al, 2009). Gene read counts were estimated using HTSeq union gene counts (Anders et al, 2015). Sequencing bias was estimated using Picard Tools (broadinstitute.github.io/picard/) functions CollectRnaSeqMetrics, CollectGcBiasMetrics, CollectAlignmentSummaryMetrics, and CollectInsertSizeMetrics. An ANOVA was used to determine if these metrics were having a significant effect on the data. Genes with expression (< 10 counts) across half the samples were excluded, leaving 17,068 genes that passed the filter for further analyses. Principle component analysis was performed in R using the function "prcomp" on data normalized by variance stabilizing transformation (VST). Differential gene expression analysis was performed using DESeq2 (Love et al, 2014) with the model ~Group + Litter. False discovery rate (FDR) < 5% was used as a cutoff to determine if genes were differentially expressed. Gene ontology and pathway analysis were performed using gprofiler2 (https://biit.cs.ut.ee/gprofiler/gost) and disgenet (https://www.disgenet.org/). This work used computational and storage services associated with the Hoffman2

Shared Cluster provided by UCLA Institute for Digital Research and Education's Research Technology Group. All data and code used in analyses will be shared publicly on GEO and github.

## Quantitative PCR

Reverse Transcriptase quantitative PCR (RT-qPCR) was performed using the SuperScript VILO First-Strand Synthesis System (Invitrogen). For each sample, >500 ng of total RNA was used for cDNA synthesis. In each qPCR reaction, 1–4 ng cDNA was combined with PowerTrack SYBR Green Master Mix (ThermoFisher) and primer pairs. All primer pairs were validated for ≥1.8 amplification efficiency. Samples were run on a ThermoFisher QuantStudio 5 real-time PCR system in triplicate, and log2 fold change was determined by normalizing the delta-delta CT to the internal reference gene Beta Actin. Primers were designed using Primer3-plus. Primer sets used; *Slc2a1* F: acttgccttctttgccaagc R: aaagcctcctagctcagagttc; *Ldha* F: aactgcaggcttcgattacc R: tgcatcatggacgtacacac; *Vegfa* F: acacgacaaacccattcctg R: tccacaaagcatgccatgtc; *Aldoa* F: ctgaataggctgcgttctcttg R: aaggactaaggagcgaacgc; *Pdk1* F: atgctggctggttttgatgc R: ttcagtcaccccgaaaatgc; *Pfkl* F: ggacaaaccgggtacacagg R: atccgcagtttctccaggtc; *Pfkfb3* F: gcatccctgagcttttgaacag R: aatgtgcttgtgccggagtc; *Slc16a4* F: tggttgtttccaccaagcag, R: taggctacatgcggagatcac; *Actb* F: tttggcgcttttgactcagg R: actttgggggatgtttgctc.

## Western blot analysis

Whole-cell extracts from mESCs cultured in feeder-free 2i media were prepared in a modified RIPA lysis buffer: 50 mM Tris (pH 7.4), 150 mM NaCl, 1% NP-40, 0.25% sodium deoxycholate, and 1x SigmaFast protease inhibitor cocktail. Spheroids were lysed in RIPA buffer and protease inhibitors. Protein concentration was determined with a BCA kit (Thermo Fisher Scientific) and normalized to 1.0–2.0 µg/µl. 20–25 µl of protein was mixed with sample buffer and 50–100 mM DTT. Protein was transferred to a PVDF membrane, blocked with either 1% BSA or Intercept (TBS) Blocking Solution (LI-COR), and blotted with primary antibodies at 4 °C overnight. Primary antibodies used: Foxp1 (1:1000, D35D10, #4402, Cell Signaling) (Glut1 (2 µg/ml, #sc-377228, Santa Cruz Biotechnology), HIF-1α (1 µg/ml, #AF1935, R&D Systems), Ldha (1:1000, #2012 Cell Signaling), Vegfa (1:500, #19003, Proteintech), Beta-Actin (1 µg/ml, 664802, Biolegend), Gapdh (1:1000, #926-422116, LI-COR), α-Tubulin (Synaptic Systems, 1:2000, #302 204). Western blots were imaged on an Odyssey DLx Imager system.

## RNA probe synthesis and in situ hybridization

Riboprobes were generated using primers designed against the 5′ or 3′ UTRs of mouse *Aldoa*, *Ldha*, *Slc16a4*, *Pfkl*, *Pfkfb3*, *Pdk1,* and *Slc2a1* transcripts. In situ hybridization was performed on sections as previously described (Pearson et al, 2011). Primer sets used: *Aldoa* F: cactggggtcactttcctgt R: aagggatggcagatttagca; *Ldha* F: ggcttctaggcagaccacac R: gacacttgggtggttgggttc; *Slc16a4* F: gaggtccagagactggcaac R: tgtcccttaggcagagatgg; *Pfkl* F: gcatcaccaacctgtgtgtc R: tgatgatgttcagccgagag; *Pdk1* F: ttctcctgcagcctacccta R: gcaccccttgtctgagcttc; *Slc2a1* F: agcagtgaagtccaggagga R: ctggtctcaggcaaggaaag; *Pfkfb3* F: cccagcttcctgtgtagagc R: agaggagtcagggcaagtca.

## RNAScope fluorescent in situ hybridization

RNAScope (acdbio) FISH was performed on cryosections per manufacturer guidelines. Probes used were Vegfa #412261-C3 and Foxp1#485221.

## Immunohistochemistry

Immunohistochemistry was performed on tissue sections as previously described (Pearson et al, 2020). In brief, slides were post-fixed for 10 min in 4% PFA (post-fix was not included when performing IHC on spheroids). After three PBST (0.1% Triton-X) washes and 10 min in antibody block, primary antibodies were applied overnight. Primary antibodies were used at the following dilutions: Foxp1 1:16,000 (Rousso et al, 2012), Glut1 1:100 (Abcam ab128033), Ldha 1:250 Abcam ab52488, HIF-1α (1:100 R&D Systems AF1935), TUJ1 1:1000 (BioLegend 801201), Vegfa 1:250 (Proteintech 19003-1-AP), Tbr2 1:500 (Millipore Sigma AB15894), CD31 1:100 (R&D Systems AF3628), Nestin 1:1000 (Novus NB100-1604), Tbr1 1:1000 (Abcam ab31940), Ctip2 1:1000 (Abcam ab18465), Pax6 1:1000 (MBL PD022). Donkey secondary antibodies (Jackson Immunoresearch) and DAPI (1mg/ml) were used at 1:1000. 2 µM of Isolectin GS-IB4 from *Griffonia simplicifolia* conjugated to Alexa Fluor 647 (ThermoFisher Cat.I32450) was diluted with secondary antibodies and incubated on slides for 1 h at room temperature. When expression levels were to be measured, coverslips were applied to ensure even distribution across the slide.

## Image acquisition

Confocal images were acquired using Zeiss LSM 780, LSM 800, and Olympus FV1000 laser scanning confocal microscopes and processed with Zen Blue and Fluoview software. DIC images of in situ hybridizations were collected using a Zeiss Axioimager microscope and Axiovision software.

## Quantification and statistical analysis

N numbers and statistical analyses for each figure are in each figure legend. Images were processed and compiled using Adobe Photoshop with adjustments applied to the entire image and restricted to brightness, contrast, and levels. Images shown in figures as comparisons, e.g., intensity levels, were obtained and processed in parallel using identical settings. Composite images were assembled using Adobe Illustrator software.

For each experiment, the mean gray values (relative to background) of labeled Nestin[+] RG per section were quantified from 3–6 sections per embryo (sampled at ~100 µm intervals along the rostrocaudal axis within the presumptive somatosensory cortex). Each mutant's mean gray value of staining was normalized to littermate controls. Mean gray values were measured using Fiji software, with background staining levels subtracted. For vessel measurements, contiguous IB4[+] vessels in 50-µm-thick sections were identified using surface rendering software in Imaris (minimum threshold 742, minimum number of voxels $1.7 \times 10^4$). Measurements for each PVP vessel were recorded (PNVP vasculature was not included). In Fig. 3A,B, each hemisphere was divided into three identically sized dorsal-ventral bins, and vessels

within each bin were quantified. Vessel length, volume, and branch points were analyzed using Imaris imaging software.

gprolifer and disgenet online database tools were used to analyze differentially expressed RNA Sequencing gene lists. Imaris was used to analyze blood vessel length and branching; surface-rendered images were generated to identify and measure individual vessels. GraphPad Prism software was used to determine the normality of each dataset (using the Shapiro–Wilk test and Kolmogorov–Smirnov Normality tests), and the appropriate parametric test was applied. Student's t-tests and Mann–Whitney tests were calculated using Prism software. Significance was assumed when $p < 0.05$. The results of statistical tests ($p$ values and sample sizes) are reported in Figure legends. All data are presented as mean ± SEM. Sample size estimates were not used. Spheroid data sets were blinded. Embryonic cortex data sets were not blinded.

## Data availability

RNA-seq data have been deposited at NCBI Gene Expression Omnibus, GEO accession number GSE217364 are publicly available as of the publication date. No original code was created. Any additional information required to reanalyze the data reported in this paper is available from the lead contact upon request. Further information and requests for resources and reagents should be directed to the Lead Contact, Caroline Alayne Pearson (cap4010@med.cornell.edu).

## Peer review information

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

## Acknowledgements

We are grateful to H. Tucker and the RIKEN BioResource Center for mouse strains used in this study. We appreciate K. Phan, A. Schlusche, and S. Singh for technical assistance, the UCLA Broad Stem Cell Research Center (BSCRC) and the Center for Neurogenetics at WCM for microscopy and other resources, and the UCLA Sequencing Core for technical assistance. We thank Novitch and Ross lab members for their valuable input and discussions, and M. Placzek, M.G Andrews, C. Iadecola, and S.J Butler for critical input on the manuscript. JEB was supported by the UCLA BSCRC Rose Hills Foundation Graduate Scholarship Training Program (EDUC4-12753). LG, TM, and JHK were supported by the NIH R00GM132518 and the NIH/NCI Cancer Center Support Grant P30CA015704. MRH was supported by R01NS126209 and the American Heart Association Career Development Award (AHA941434). NIH R01NS105477 supported MER. The UCLA Broad Stem Cell Research Center and the NIH R01NS089817 supported BGN. CAP was supported by research awards from the UCLA-California Institute for Regenerative Medicine Training Grant (TG2-01169), the Brain and Spine Institute at New York Presbyterian Hospital and Weill Cornell Medicine, the GLUT1-Deficiency Foundation and the American Epilepsy Society Junior Investigator award. We also acknowledge the support of the NINDS-sponsored UCLA Informatics Center for Neurogenetics and Neurogenomics (P30NS062691) and the UCLA Intellectual and Developmental Disabilities Research Center Genetics and Genomics Core supported by the NICHD (U54HD087101 and P50HD103557).

## Author contributions

**Jessie E Buth**: Formal analysis; Investigation; Methodology; Writing—original draft. **Catherine E Dyevich**: Formal analysis; Investigation; Methodology; Writing—original draft; Writing—review and editing. **Alexandra Rubin**: Formal analysis; Investigation; Methodology; Writing—review and editing. **Chengbing Wang**: Investigation; Methodology. **Lei Gao**: Investigation; Methodology. **Tessa Marks**: Investigation; Methodology. **Michael RM Harrison**: Conceptualization; Formal analysis; Investigation; Methodology. **Jennifer H Kong**: Resources; Funding acquisition; Investigation; Methodology. **M Elizabeth Ross**: Resources; Funding acquisition; Investigation; Methodology; Writing—original draft; Writing—review and editing. **Bennett G Novitch**: Conceptualization; Resources; Funding acquisition; Methodology; Writing—original draft; Project administration; Writing—review and editing. **Caroline Alayne Pearson**: Conceptualization; Resources; Data curation; Formal analysis; Supervision;

Funding acquisition; Validation; Investigation; Visualization; Methodology; Writing—original draft; Project administration; Writing—review and editing.

## Disclosure and competing interests statement

The authors declare no competing interests.

# Expanded View Figures

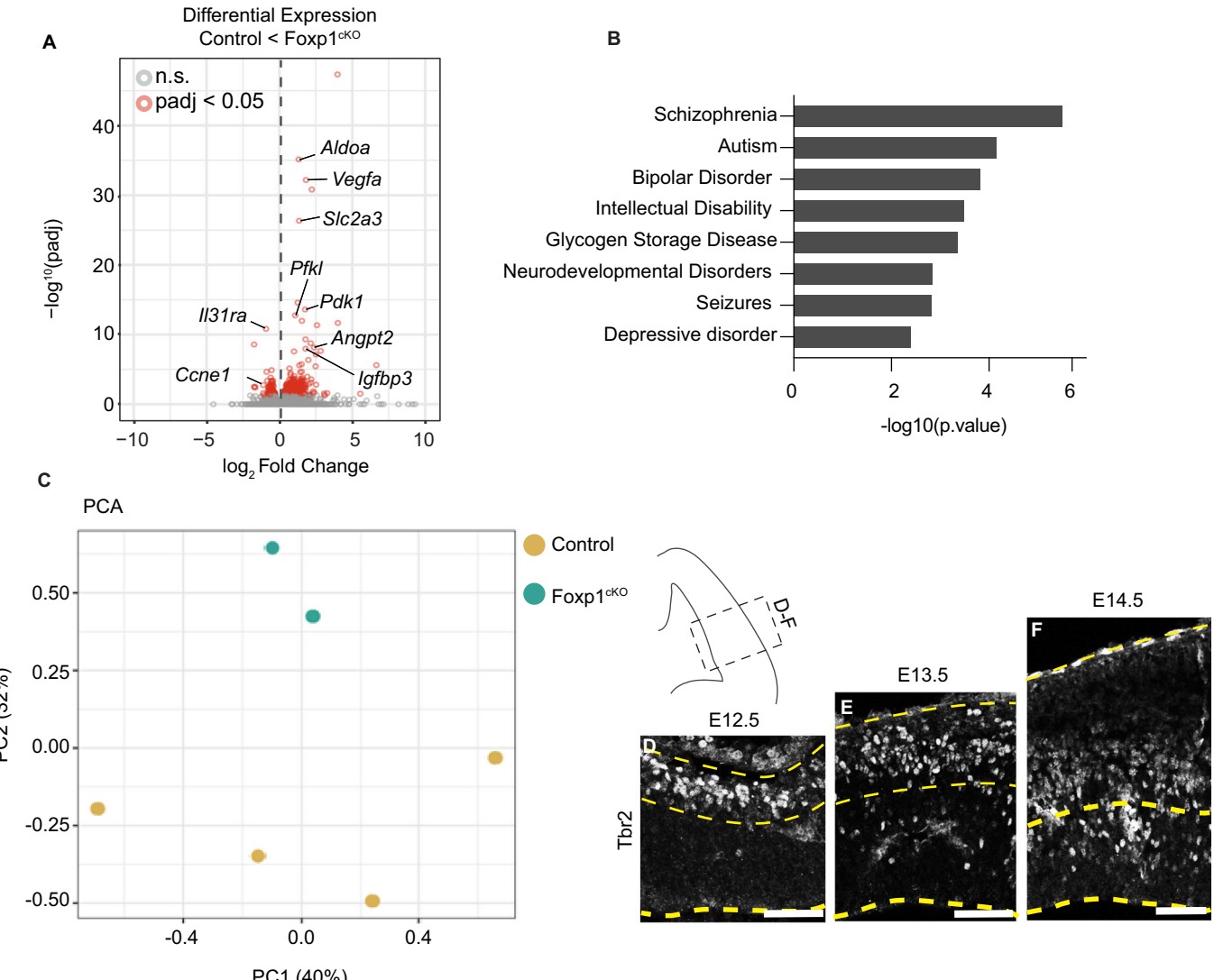

**Figure EV1. RNA Seq analysis of Foxp1cKO cortex.**

(A) Volcano plot of gene expression changes in the absence of Foxp1 in E12.5 lateral cortex compared to control embryos. Gray circles denote non-significant gene changes (adjusted *p*-value > 0.05); red circles denote significantly differentially expressed genes (adjusted *p*-value < 0.05). (B) Human disorders associated with genes significantly misregulated in Foxp1cKO mutants at E12.5. (C) The principal component analysis (PCA) of control and Foxp1cKO mutants shows PC1 and PC2. (D–F) IHC for Tbr2+ intermediate progenitors in wild-type cortex at E12.5, E13.5, and E14.5. Schematic denotes area image in (D–F). Scale bars 50 µm. Data information: significance determined by ANOVA (A, B).

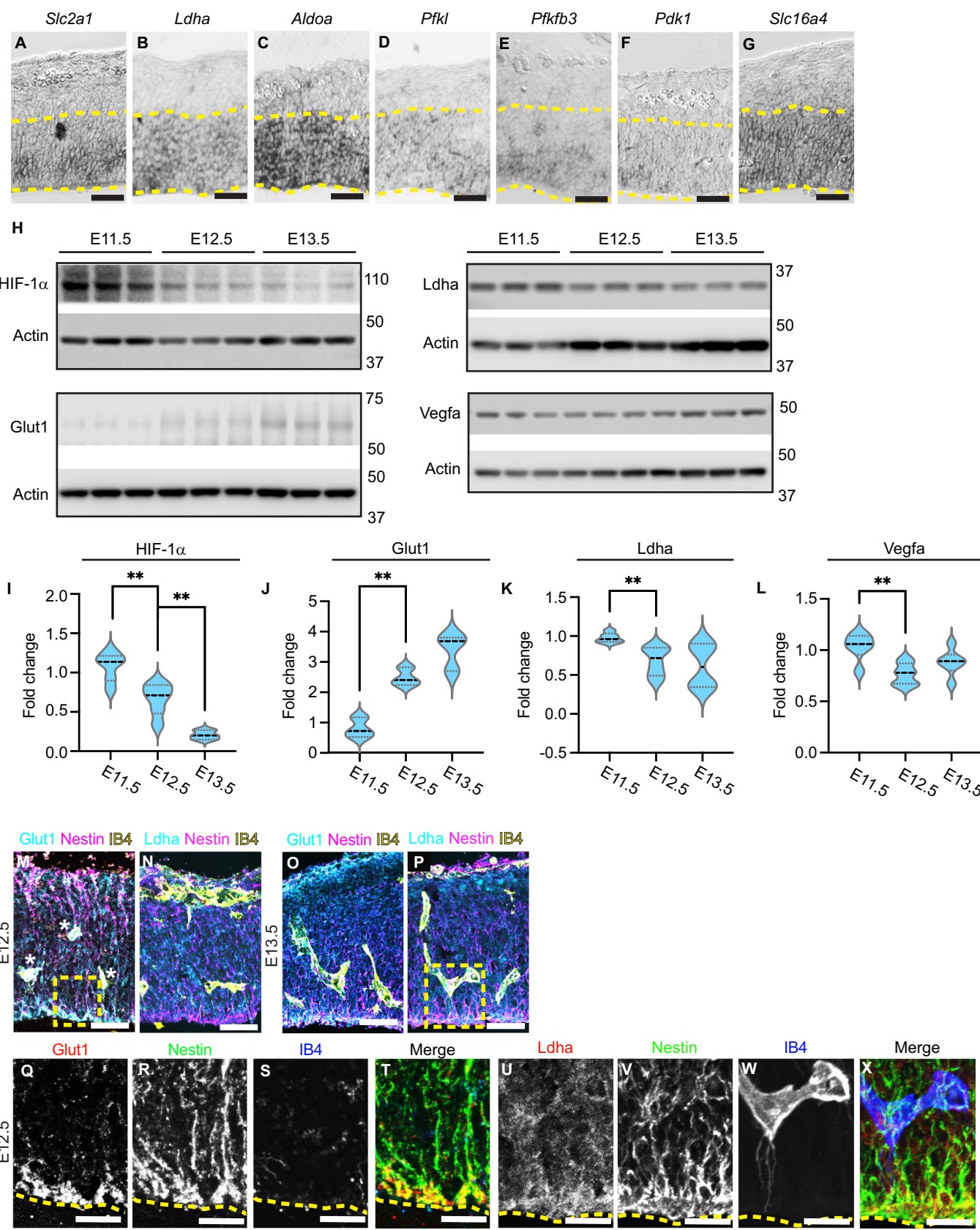

**Figure EV2. HIF-1α target gene expression in RG in the wild-type cortex.**

(A–G) Wild-type mRNA expression of glycolysis genes *Slc2a1, Ldha, Aldoa, Pfkl, Pfkfb3, Pdk1,* and *Slc16a4* in the wild-type lateral cortex at E12.5. (H) Western blot analysis of HIF-1α, Glut1, Ldha, and Vegfa (with Beta Actin) in wild-type cortical lysates at E11.5, E12.5, E13.5. (I) Fold change of HIF-1α levels normalized to Beta Actin between E11.5 and E13.5. (J) Fold change of Glut1 levels normalized to Beta Actin between E11.5 and E13.5. (K) Fold change of Ldha levels normalized to Beta Actin between E11.5 and E13.5. (L) Fold change of Vegfa levels normalized to Actin between E11.5 and E13.5. (M, N) IHC for Glut1 and Ldha with Isolectin B4, and Nestin at E12.5 in the wild-type cortex. (O, P) IHC for Glut1 or Ldha with Isolectin B4, and Nestin at E13.5 in the wild-type cortex. Boxed areas are magnified in (Q–X). (Q–T) High magnification image of IHC for Glut1 in Nestin+ RG at E12.5. Isolectin B4 labels blood vessels. (U–X) High magnification images of IHC for Ldha in Nestin+ RG at E13.5. Isolectin B4 labels blood vessels. Scale bars 50 μm (A–K), 10 μm (L–S). Data information: $N = 5$–9 embryos per time point, 3–6 replicates. $p = 0.0039$ and 0.0073 (I). $N = 5$–9 embryos per time point, 3 replicates. $p = 0.0028$ (J). $N = 5$–9 embryos per time point, 6 replicates. $p = 0.0041$ (K). $N = 5$–9 embryos per time point, 6 replicates. $p = 0.0017$ (L). All Student's t-tests. All data represented as mean ± SEM. Source data are available online for this figure.

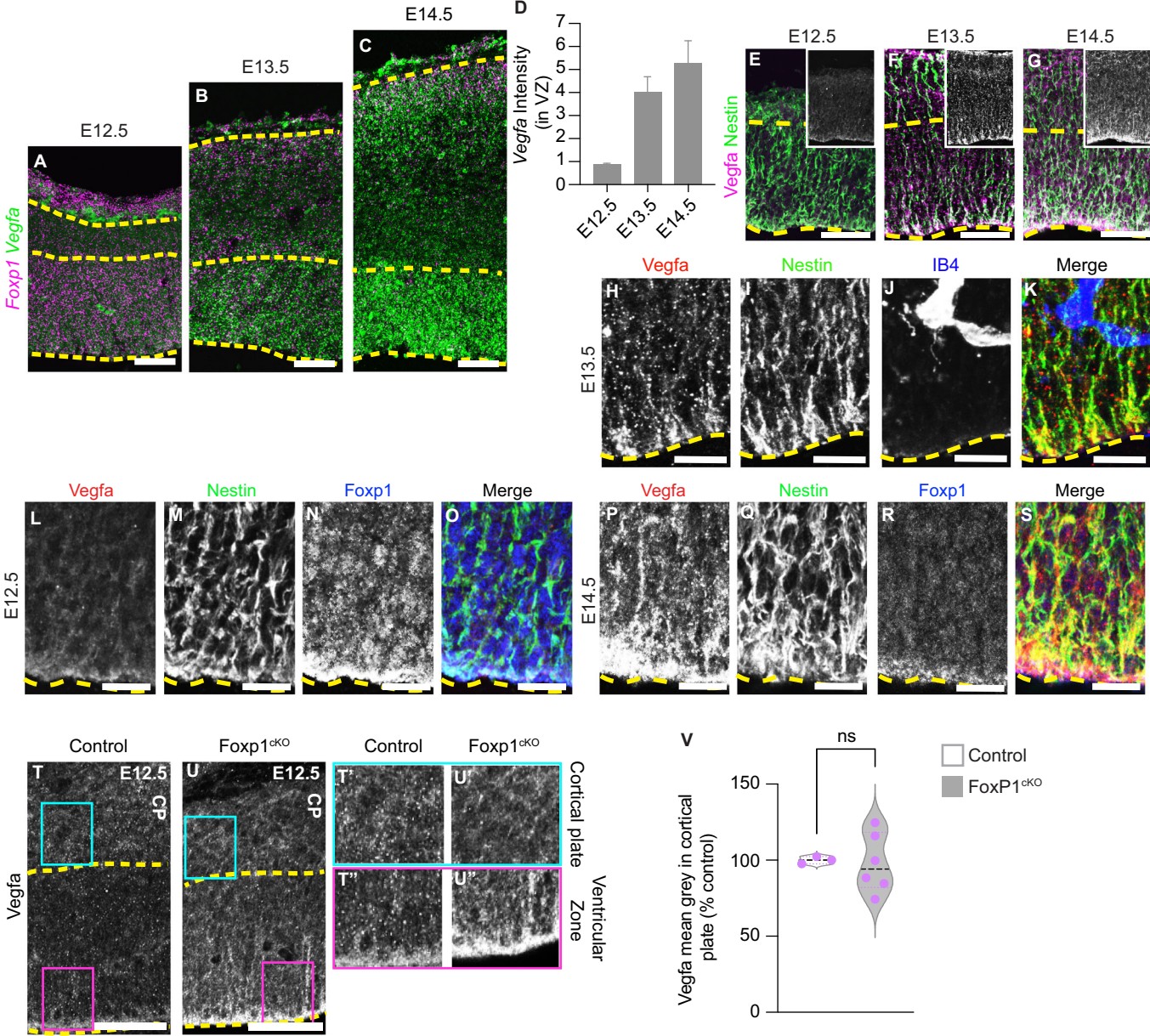

**Figure EV3. Spatial and temporal relationship between Foxp1 and Vegfa in radial glia and neurons.**

(A–C) RNA Scope analysis of Vegfa and Foxp1 in the wild-type cortex at E12.5–E14.5. (D) *Vegfa* mean gray value (over background) in the ventricular zone between E12.5–E14.5. (E–G) IHC for Vegfa and Nestin in the wild-type cortex at E12.5, E13.5, and E14.5. (H–K) High magnification images of Vegfa, Nestin, and IB4 at the apical surface of the VZ at E13.5. (L–S) High magnification images of IHC for Vegfa, Nestin, and Foxp1 at E12.5 and E14.5 in wild-type cortex. (T–U) IHC for Vegfa in the cortex at E12.5 in control and Foxp1cKO embryos. Cyan boxes represent areas in CP magnified in **T**,′ and **U**.′ Magenta boxes represent the area within VZ magnified in **T″** and **U″**. (V) Quantification of Vegfa mean gray values in the cortical plate in control and Foxp1cKO cortex at E12.5. Scale bars 50 μm (**A–G**, **T**, **U**) 10 μm (**H–S**). Data information: N = 5–7 embryos per time point (**D**). N = 3 control, 6 mutant (2 litters). p = 0.8674, Student's t-test (**V**). All data represented as mean ± SEM.

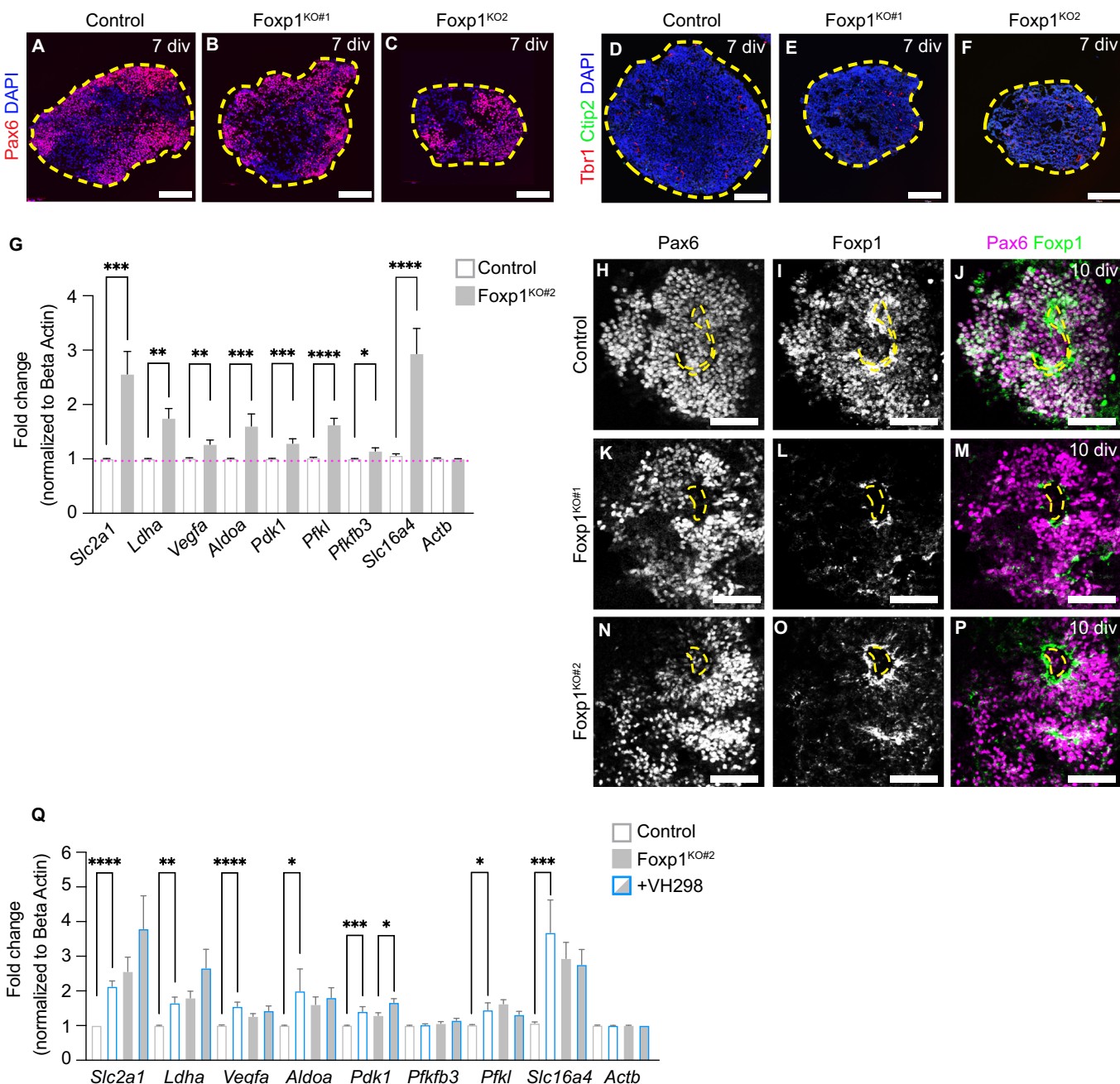

**Figure EV4.  Analysis of Foxp1-deficient cortical spheroids.**

(A–C) IHC for Pax6 in control and Foxp1$^{KO}$ spheroids at 7 days in vitro (div). (D–F) IHC for Ctip2 and Tbr1$^+$ neurons in control and Foxp1$^{KO}$ spheroids at 7 div. (G) qPCR analysis of HIF-1α target gene expression in control and Foxp1$^{KO\#2}$ spheroids at 10 div. (H–P) IHC for Foxp1 in Pax6$^+$ NPCs in control and Foxp1$^{KO}$ spheroids at 10 div. (Q) qPCR analysis of HIF-1α target gene expression in control and Foxp1$^{KO\#2}$ spheroids at 10 div. treated with DMSO or VH298. Scale bars 100 µm (A–F), 50 µm (H–P). Data information: $N = 10$–$12$ spheroids from 3 individual batches. $p = 0.0007$ (*Slc2a1*), 0.0108 (*Ldha*), 0.0031 (*Vegfa*), 0.0009 (*Aldoa*), 0.0002 (*Pdk1*), 0.0002 (*Pfkl*), 0.0226 (*Pfkfb3*), <0.0001 (*Slc16a4*). Student's t-test (G). $p = $ <0.0001 (*Slc2a1*, control), 0.0013 (*Ldha*, control) <0.0001 (*Vegfa*, control), 0.0179 (*Aldoa*, control) 0.0002 (*Pdk1*, control), 0.0312 (*Pdk*1, Foxp1$^{KO\#2}$), 0.0201 (*Pfkl*, control), 0.0003 (*Slc16a4*, control). Student's t-test (Q). All data represented as mean ± SEM.

