## [Peer Review File · EMBO Reports]

Foxp1 suppresses cortical angiogenesis and attenuates HIF-1alpha signaling to promote neural progenitor cell maintenance.

Jessie Buth, Catherine Dyeovich, Alexandra Rubin, Chengbing Wang, Lei Gao, Tessa Marks, Michael Harrison, Jennifer Kong, M. Elizabeth Ross, Bennett Novitch, and Caroline Pearson

Corresponding author(s): Caroline Pearson (cap4010@med.cornell.edu)

Review Timeline:

Submission Date:	5th Jun 23
Editorial Decision:	5th Jul 23
Revision Received:	8th Feb 24
Editorial Decision:	13th Mar 24
Revision Received:	16th Mar 24
Accepted:	22nd Mar 24

Editor: Esther Schnapp

Transaction Report:

Dear Dr. Pearson,

Thank you for the submission of your manuscript to EMBO reports. We have now received the full set of referee reports that is pasted below.

As you will see, the referees acknowledge that the findings are potentially interesting. However, they also all raise several and also overlapping concerns. I think all concerns are reasonable and should be addressed. Please let me know in case you disagree, and we can discuss the exact revision requirements further, also in a video chat if you like.

I would thus like to invite you to revise your manuscript with the understanding that the referee concerns must be fully addressed and their suggestions taken on board. Please address all referee concerns in a complete point-by-point response. Acceptance of the manuscript will depend on a positive outcome of a second round of review. It is EMBO reports policy to allow a single round of major revision only and acceptance or rejection of the manuscript will therefore depend on the completeness of your responses included in the next, final version of the manuscript.

We realize that it is difficult to revise to a specific deadline. In the interest of protecting the conceptual advance provided by the work, we recommend a revision within 3 months (5th Oct 2023). Please discuss the revision progress ahead of this time with the editor if you require more time to complete the revisions.

You can either publish the study as a short report or as a full article. For short reports, the revised manuscript should not exceed 27,000 characters (including spaces but excluding materials & methods and references) and 5 main plus 5 expanded view figures. The results and discussion sections must further be combined, which will help to shorten the manuscript text by eliminating some redundancy that is inevitable when discussing the same experiments twice. For a normal article there are no length limitations, but it should have more than 5 main figures and the results and discussion sections must be separate. In both cases, the entire materials and methods should be included in the main manuscript file.

- 1) A data availability section providing access to data deposited in public databases is missing. If you have not deposited any data, please add a sentence to the data availability section that explains that.
- 2) Your manuscript contains statistics and error bars based on $n=2$. Please use scatter blots in these cases. No statistics should be calculated if $n=2$.

5) a complete author checklist, which you can download from our author guidelines . Please insert information in the checklist that is also reflected in the manuscript. The completed author checklist will also be part of the RPF.

6) Please note that all corresponding authors are required to supply an ORCID ID for their name upon submission of a revised

manuscript (). Please find instructions on how to link your ORCID ID to your account in our manuscript tracking system in our Author guidelines

- the name of the statistical test used to generate error bars and P values,
- the number (n) of independent experiments (please specify technical or biological replicates) underlying each data point,
- the nature of the bars and error bars (s.d., s.e.m.),
- If the data are obtained from n {less than or equal to} 2, use scatter blots showing the individual data points.

I look forward to seeing a revised form of your manuscript when it is ready.

Yours sincerely,

Referee #1:

In this manuscript, Buth et al. investigate cellular and molecular pathways triggered by the autism-associated transcriptional repressor Foxp1 during corticogenesis. They report that Foxp1 expression in cortical progenitors inhibits angiogenesis via repressing HIF1a-target genes.

The authors previously described that Foxp1 promotes self-renewal in cortical radial glia cells as opposed to neuronal differentiation (Pearson et al 2020). They now follow up on Foxp1 downstream molecular mechanisms in the neurogenesis context. Their results revealed that several genes upregulated upon Foxp1 deletion are HIF1a targets. Consequently, mutant mice lacking Foxp1 in cortical progenitors (Emx1+) displayed angiogenesis-related phenotypes in the neocortex. They also identified other differentially expressed genes related to glycolysis but did not follow up on this issue.

Conceptually, the finding that a gene expressed in neural stem cells affects the vascularization of the neocortex is relevant for the neurodevelopment field, as only a few examples of this kind have been found so far. Additionally, Foxp1 has a well-documented role in cognitive functions and is associated with psychiatric conditions, providing an attractive angle to this story and adding indirect evidence to the emerging idea that vascular alterations might contribute to the etiology of neurodevelopmental disorders. I believe that the conclusions regarding Foxp1 involvement in embryonic brain angiogenesis are well-drawn and the data is convincing. The manuscript is well-written and easy to follow.

1. I have, however, a significant concern related to a main conclusion mentioned throughout the paper. The authors assign a causal link between the previously-described role of Foxp1 in neural stem cell fate decisions and Foxp1-induced angiogenesis-related alterations (summarized in Figure 4 N,O). This link is not evaluated in the manuscript; the two datasets are only correlative, and thus, this conclusion is too speculative. If the authors provide evidence for a causal link, the impact of their study will be certainly enhanced.

The authors could try to rescue Foxp1-induced vascular alterations in Foxp1 cKO mice and analyze whether Foxp1 deletion still affects stem cell fate decisions. I understand that a genetic approach would take too long, but other experiments could also help elucidate this issue. For example, the authors could downregulate HIF1a activity in Foxp1 cKO embryos by exposing pregnant dams to decreased oxygen levels (as in Lange et al 2016). Alternatively, they could perform in utero electroporations to knockdown/knockout Vegfa in Foxp1 cKO embryos.

Another experiment to understand the contribution of the microenvironment on Foxp1-neural effects is to isolate cortical progenitors from E12.5 wild type and Foxp1 cKO mice and test their fate decisions in vitro. This experiment would not discard an autocrine VEGFA-mediated effect -in case the fate decision effects are maintained- but it would rule out a vessel-mediated effect. In addition, the expression of HIF1a target genes could also be easily manipulated in the in vitro setting.

Other suggestions:

2. It would be interesting to know how the brain vasculature develops in Foxp1 cKO mice. Whether the observed alterations at E13.5 are compensated over time, or if there is still a vascular phenotype at later time points. This additional information is easy to obtain and is important to better illustrate their main finding and get a bigger picture.

Furthermore, this might be relevant to speculate on whether acute/chronic mild alterations in brain vasculature could have long-term effects, although I recognize that we still do not know what is the contribution of Foxp1 cKO vascular phenotypes to brain functional alterations observed at later stages (as described in Araujo et al., 2017).

3. Please provide an immunostaining of Foxp1 in Foxp1 cKO brains to confirm the specific Foxp1 deletion at the developmental window used in the study.

4. I acknowledge the value of immunohistochemistry to estimate gene expression variations in a cell-specific manner. However, immunostainings are not truly quantitative. Although the authors provide complementary qPCR measurements in Fig S3, they do it only for some of the genes depicted in their work. Since they already have the material, I suggest that they complete the dataset including all the genes of interest.

5. As Foxp1 is also highly expressed in endothelial cells from the developing brain (UCSC Cell Browser: "cortex development" and "vasculature in the developing brain" databases) and a cell-autonomous role of Foxp1 in angiogenesis has been described (Grundmann 2013), I suggest that the authors take these data into account when discussing possible impacts of Foxp1 mutations in humans (last paragraph of the article, line 230).

6. Supplementary Fig S4 nicely supports the finding that Foxp1 deletion in cortical progenitors induces angiogenesis via Vegfa upregulation. I suggest to include this panel in main Figure 4.

7. RNAseq dataset design is not optimal, with only 2 samples from KO vs 4 control. Why does Foxp1 not appear as a downregulated gene in the DEG dataset?

8. Fig.2: The difference in medial vs. dorsal HIF1a expression illustrated in Fig.2 M is not obvious from the picture (Fig.2 I).

9. Scale bars are missing in some pictures and figure legends.

I understand that the article is in a Report format. Nevertheless, additional information could be added by replacing some panels in Figures 1 and 2 that could go to supplementary information (Fig. 1 E-J is not a novel finding, and expression patterns in wild type brains from Fig. 2 could also be a supplementary figure).

Referee #2:

The manuscript by Buth et al., entitled "Foxp1 suppresses cortical angiogenesis and attenuates HIF1alpha signaling in neural progenitor cells", provides evidence that conditional KO Foxp1 in EMX1-Cre+ cells resulted in an increase in IB4+ blood vessels, which associated with the elevated expressions of HIF1alpha and its target gene (e.g., VEGFa) expression. While this manuscript is of interest to the field, the following concerns remain to be addressed.

Major concern:

1. EMX1-Cre mouse line is known to express Cre in both excitatory neurons and glia [astrocytes and radial glia (RG)] in both cortex and hippocampus. Allen's RNAseq data base suggests that Foxp1 is expressed in both excitatory neurons and glial cells. The IHC data in Fig. 2 also showed that Foxp1 was not only enriched in the VZ, but also expressed in neurons. Thus, the conclusion of "early loss of Foxp1 in RG-radial glia", based on the data from Foxp1f/f;EMX1-Cre embryos (at E12.5), needs additional evidence. How about the phenotypes in Floxp1 CKO in embryonic neurons?
2. Can inhibition of HIF1a signaling (or VEGFa) in RG diminish or abolish the phenotype in Foxp1f/f;EMX1-Cre mice?

Other concerns:

1. Fig. 1. Please include the quantification data of Foxp1 and TUJ1 expression levels during embryonic cortical brain development (E12.5 to E14.5) to demonstrate their relationships with blood vessel development; How about other blood vessel markers (e.g., CD31), in addition to IB4?
2. Fig. 2. In addition to the IHC, please verify the Foxp1, HIF1a, and its target genes' expression by western blots.
3. Fig. 3. Again, additional methods (e.g., western blots) are necessary to verify the expression levels of HIF1a's target genes.
4. Fig. 4. As IB4 labels additional cell types in addition to blood vessels, it is necessary to immunostain other markers for blood vessels.

Referee #3:

The manuscript by Buth et al. identify a role for the transcription factor FoxP1 in the regulation of cortical angiogenesis during neurogenesis via the regulation of HIF1alpha and VEGFA signaling in neural progenitors.

The role of FoxP1 in regulating the switch between early and late cortical neurons progenitors has been previously established by the authors. The current manuscript nicely demonstrates the role of FoxP1 in transcriptome profiles modification associated with this switch. In particular, the description of genes and regulators of cell metabolism and growth as well as the communication with the extracellular environment is physiologically and pathologically relevant.

The manuscript is well written. Data mostly supports the conclusion, and the regulation of target genes expression, in particular HIF-1alpha and VEGFA, in neural progenitors in response to FoxP1 activity is convincing. The main concern regarding this study is the lack of functional evidence between the expression of VEGF-A in neural progenitors and the increase angiogenesis in the absence of FoxP1 activity. Many other cell types could potentially express VEGFA and be impacted in a non-cell autonomous manner by the depletion of FoxP1. To clarify, while the description of the phenotype supports the hypothesis of a potential functional relevance of VEGFA derived from late-neural progenitors in the angiogenic process during cortical development, it does not demonstrate it.

1-The authors mentioned (page 3, line 73): "Nevertheless, the developmental factors that coordinate angiogenesis with neurogenesis remain poorly defined".

This statement is not accurate. Previous publications reported the role of VEGFA during neurogenesis and angiogenesis coordination. Some of them associated VEGFA expression from neural progenitors, neurons and/or astrocytes with coupling between neurogenesis and angiogenesis. While astrocytes are not relevant in this study because of the developmental timing, newborn neurons or other cell types could be considered.

2-Fig2T-W and X. While I am open to believe it. It is not clear with the pictures and magnification provided that the expression VEGFA colocalizes mainly with the nestin+ cells. Opening the possibly for other cell populations expressing VEGFA at later

stages.

Similar observation for Fig. 3 data quantification. For fig2 and 3, if my understanding is correct, intensity of the staining is quantified in the region of interest (not specifically in nestin+ cells). Would that be possible to quantify expression (mRNA and/or protein) specifically in neural progenitors?

3-VEGF-A has been shown to be expressed in different types of neurons, including newborn cortical neurons. If depletion of FoxP1 promotes the neurogenic division of neural progenitors. How the increase angiogenic phenotype observed after depletion of FoxP1 is a direct consequence of VEGFA expression in neural progenitors or an indirect result of the increase in neuronal VEGF-A expression because an increased number of differentiated neurons is unclear. Neuronal VEGF-A expression should be analyzed in the different conditions of FoxP1 expression.

4-The description of VEGFA expression and regulation in neural progenitors is not completely novel. However, the concept of a switch in VEGFA expression associated with a late type of neural progenitors committed to neuronal differentiation is interesting. Conditional VEGFA expression in neural progenitors before the switch from early to late neural progenitors could demonstrate the role of VEGF-A expression in progenitors during cortical angiogenesis.

5-The contribution of VEGF-A derived from late-neural progenitors could also be functionally investigated. Conditional deletion of VEGF-A in neural progenitors derived from radial glia will help draw a more definitive conclusion regarding the role of VEGFA expression in late neural progenitors in the coordination between neurogenesis and angiogenesis.

6-Foxp1 is a target of microRNA-9. miR-9 has been shown as coupling neurogenesis and angiogenesis by the inhibition of factors regulating VEGFA expression. Potential miR-9 expression in radial glia, early and late progenitors in the cortex, as well as a putative miR-9 dependent regulation of FoxP1 during early to late progenitor identity switch could be discussed.

7-The title of the manuscript indicates that FoxP1 suppresses cortical angiogenesis. The data supports this statement. But Foxp1 has also been associated with promotion of the angiogenesis process. While a context dependent function of FoxP1 could explain this difference, this should be mentioned and discussed by the authors.

Reviewers Reports and Responses.

We thank the reviewers for their thoughtful comments and helpful suggestions. All three expressed enthusiasm for our findings linking the autism-associated transcription factor Foxp1 to the control of HIF-1 α signaling, angiogenesis, and neural progenitor maintenance in the developing cerebral cortex, and stressed their importance for the neurodevelopmental field. However, each review also raised concerns about the mechanisms and autonomy of effects, which were difficult to pick apart with our in vivo experiments. To address this challenge, we developed an alternative approach to examining Foxp1 function in neural progenitors in the absence of vascularization using in vitro differentiation of cortical spheroids derived from control and Foxp1 mutant mouse embryonic stem cell lines. These new studies provide corroboratory evidence that Foxp1 loss leads to changes in HIF-1 α protein levels, expression of some HIF-1 α target genes, and premature differentiation of neural progenitors, much like what is seen with conditional deletion of Foxp1 from neural progenitors in vivo. We were further able to leverage this in vitro platform to probe the connections between HIF-1 α signaling and progenitor behaviors by adding a HIF-1 α activating drug to Foxp1 mutant spheroids, which brought the balance between neural progenitors and differentiated neurons back to control levels. We hope that the addition of these new experiments along with our responses to the reviewers' individual comments have addressed their concerns.

Reviewer comments are in *black italics*; our responses are in plain blue text.

Referee #1:

"In this manuscript, Buth et al. investigate cellular and molecular pathways triggered by the autism-associated transcriptional repressor Foxp1 during corticogenesis. They report that Foxp1 expression in cortical progenitors inhibits angiogenesis via repressing HIF1 α -target genes.

The authors previously described that Foxp1 promotes self-renewal in cortical radial glia cells as opposed to neuronal differentiation (Pearson et al 2020). They now follow up on Foxp1 downstream molecular mechanisms in the neurogenesis context. Their results revealed that several genes upregulated upon Foxp1 deletion are HIF1 α targets. Consequently, mutant mice lacking Foxp1 in cortical progenitors (Emx1⁺) displayed angiogenesis-related phenotypes in the neocortex. They also identified other differentially expressed genes related to glycolysis but did not follow up on this issue.

Conceptually, the finding that a gene expressed in neural stem cells affects the vascularization of the neocortex is relevant for the neurodevelopment field, as only a few examples of this kind have been found so far. Additionally, Foxp1 has a well-documented role in cognitive functions and is associated with psychiatric conditions, providing an attractive angle to this story and adding indirect evidence to the emerging idea that vascular alterations might contribute to the etiology of neurodevelopmental disorders. I believe that the conclusions regarding Foxp1 involvement in embryonic brain angiogenesis are well-drawn and the data is convincing. The manuscript is well-written and easy to follow.

1. I have, however, a significant concern related to a main conclusion mentioned throughout the paper. The authors assign a

causal link between the previously-described role of Foxp1 in neural stem cell fate decisions and Foxp1-induced angiogenesis-related alterations (summarized in Figure 4 N,O). This link is not evaluated in the manuscript; the two datasets are only correlative, and thus, this conclusion is too speculative. If the authors provide evidence for a causal link, the impact of their study will be certainly enhanced.

The authors could try to rescue Foxp1-induced vascular alterations in Foxp1 cKO mice and analyze whether Foxp1 deletion still affects stem cell fate decisions. I understand that a genetic approach would take too long, but other experiments could also help elucidate this issue. For example, the authors could downregulate HIF1a activity in Foxp1 cKO embryos by exposing pregnant dams to decreased oxygen levels (as in Lange et al 2016)."

We thank the reviewer for their suggestion. We decided that an *in vitro* approach would be the most reliable way to perform rescue experiments and provide evidence of a causal link. Therefore, we generated two Foxp1 knock-out mouse embryonic stem cell lines, differentiated them into cortical RG-like neural progenitors (as spheroids), and conducted a variety of analyses showcased in our revised Figure 4. These analyses show that Foxp1^{KO} spheroids recapitulate several phenotypes seen in Foxp1^{cKO} cortices *in vivo*, namely premature neural progenitor differentiation, reduced HIF-1 α protein levels, and increased expression of genes that act downstream of HIF-1 α signaling. We then performed experiments culturing control and Foxp1^{KO} spheroids in the presence of VH298 – a drug that blocks HIF-1 α degradation, or DMSO (the solvent in which VH298 is dissolved). These experiments demonstrated the rescue of Foxp1-related phenotypes, i.e., stabilization of HIF-1 α protein levels, a significant increase in NPCs to control levels, and a concomitant decrease in neurons. These additional data significantly strengthen our conclusions that Foxp1 acts to attenuate HIF-1 α pathway signaling and thereby promote NPC maintenance.

"Alternatively, they could perform in utero electroporations to knockdown/knockout Vegfa in Foxp1 cKO embryos."

We appreciate the reviewer's suggestion. We decided that this alternative would not be feasible as this would require electroporation earlier than E12.5 to efficiently knockdown Vegfa before loss of Foxp1. This is technically very challenging given the low viability of embryos electroporated before E12.5.

"Another experiment to understand the contribution of the microenvironment on Foxp1-neural effects is to isolate cortical progenitors from E12.5 wild type and Foxp1 cKO mice and test their fate decisions in vitro. This experiment would not discard an autocrine VEGFA-mediated effect -in case the fate decision effects are maintained- but it would rule out a vessel-mediated effect."

As mentioned above, we conducted experiments using Foxp1^{KO} mESCs to generate cortical NPCs and observed the same increases in HIF1- α target gene expression and downregulation of HIF-1 α protein expression that we see *in vivo*. In addition, we found that Foxp1^{KO} spheroids displayed a significant decrease in Pax6⁺ NPCs and an increase in Ctip2⁺ neurons relative to

control spheroids. Thus, the phenotypes we report *in vivo* are recapitulated *in vitro*. This demonstrates a cell-autonomous role for Foxp1 in attenuating the HIF-1 α signaling pathway, including Vegfa expression.

“In addition, the expression of HIF1 α target genes could also be easily manipulated in the in vitro setting.”

We appreciate this comment, as this was the route that we decided to take to provide evidence of a causal link between the effects of Foxp1 loss on HIF-1 α signaling and progenitor maintenance.

“Other suggestions:

2. It would be interesting to know how the brain vasculature develops in Foxp1 cKO mice. Whether the observed alterations at E13.5 are compensated over time, or if there is still a vascular phenotype at later time points. This additional information is easy to obtain and is important to better illustrate their main finding and get a bigger picture.”

Given the brief nature of our report, we have decided that a thorough characterization of brain vascularization goes beyond the scope of this present study.

“Furthermore, this might be relevant to speculate on whether acute/chronic mild alterations in brain vasculature could have long-term effects, although I recognize that we still do not know what is the contribution of Foxp1 cKO vascular phenotypes to brain functional alterations observed at later stages (as described in Araujo et al., 2017).”

We thank the reviewer for this comment and have included this point in our revised results/discussion section.

“3. Please provide an immunostaining of Foxp1 in Foxp1 cKO brains to confirm the specific Foxp1 deletion at the developmental window used in the study.”

We have published the timing of Cre recombination and loss of Foxp1 protein in our previous Cell Reports article (Pearson et al. *Cell Rep* 2020) and refer readers to these earlier results.

“4. I acknowledge the value of immunohistochemistry to estimate gene expression variations in a cell-specific manner. However, immunostainings are not truly quantitative. Although the authors provide complementary qPCR measurements in Fig S3, they do it only for some of the genes depicted in their work. Since they already have the material, I suggest that they complete the dataset including all the genes of interest.”

We have added other genes of interest to our qPCR data. We have also found that most of these transcriptional changes are recapitulated *in vitro*.

“5. As Foxp1 is also highly expressed in endothelial cells from the developing brain (UCSC Cell Browser: "cortex development" and "vasculature in the developing brain" databases) and a cell-autonomous role of Foxp1 in angiogenesis has been described (Grundmann 2013), I suggest that the authors take these data into account when discussing possible impacts of Foxp1 mutations in humans (last paragraph of the article, line 230).”

We appreciate this suggestion and have made modifications to the text to include this point in our discussion.

“6. Supplementary Fig S4 nicely supports the finding that Foxp1 deletion in cortical progenitors induces angiogenesis via Vegfa upregulation. I suggest to include this panel in main Figure 4.”

As suggested, we have moved this panel into our revised main Figure 3, describing the vasculature phenotype in the Foxp1cKO cortex.

“7. RNAseq dataset design is not optimal, with only 2 samples from KO vs 4 control. Why does Foxp1 not appear as a downregulated gene in the DEG dataset?”

During the processing of our RNA-Seq dataset, we were required to remove the male samples as one of them was a significant outlier. Therefore, we have 2 mutant and 4 control female samples. With this in mind, we have confirmed transcriptional changes with qPCR, focusing on transcripts involved in glycolysis, angiogenesis, and HIF-1 α signaling.

The strategy employed in the design of the Foxp1 floxed mice only removes exons 11 and 12, as this region encodes the forkhead domain. Thus, Foxp1 transcripts are still present, but we have shown in our 2020 Cell Reports paper that Foxp1 protein is no longer detected at E11.5 after Cre recombination.

“8. Fig.2: The difference in medial vs. dorsal HIF1a expression illustrated in Fig.2 M is not obvious from the picture (Fig.2 I).”

This figure has been rearranged and now shows the medial vs. dorsal differences more clearly.

“9. Scale bars are missing in some pictures and figure legends.”

We have added missing scale bars. as suggested.

“I understand that the article is in a Report format. Nevertheless, additional information could be added by replacing some panels in Figures 1 and 2 that could go to supplementary information (Fig. 1 E-J is not a novel finding, and expression patterns

in wild type brains from Fig. 2 could also be a supplementary figure).”

We appreciate the reviewer’s suggestion and accordingly combined Figures 2 and 3 from the previous submission and moved most of the wild-type analyses to Expanded View Figures EV2 and EV3.

Referee #2:

The manuscript by Buth et al., entitled "Foxp1 suppresses cortical angiogenesis and attenuates HIF1alpha signaling in neural progenitor cells", provides evidence that conditional KO Foxp1 in EMX1-Cre+ cells resulted in an increase in IB4+ blood vessels, which associated with the elevated expressions of HIF1alpha and its target gene (e.g., VEGFa) expression. While this manuscript is of interest to the field, the following concerns remain to be addressed.

Major concern:

1. EMX1-Cre mouse line is known to express Cre in both excitatory neurons and glia [astrocytes and radial glia (RG)] in both cortex and hippocampus. Allen's RNAseq data base suggests that Foxp1 is expressed in both excitatory neurons and glial cells. The IHC data in Fig. 2 also showed that Foxp1 was not only enriched in the VZ, but also expressed in neurons. Thus, the conclusion of "early loss of Foxp1 in RG-radial glia", based on the data from Foxp1^{ff};EMX1-Cre embryos (at E12.5), needs additional evidence. How about the phenotypes in Floxp1 CKO in embryonic neurons?"

We thank the reviewer for this comment. To address the relationship between Foxp1 and Vegfa in neurons, we have analyzed the expression levels of Vegfa in neurons in several ways. First, we have analyzed the co-expression of Vegfa and Foxp1 in neurons at mid- and late-neurogenic stages (see new Expanded View Figure EV3). This analysis shows that in neurons, Vegfa and Foxp1 are coexpressed in neurons and don’t show the same inverse relationship that we describe in RG. We have also analyzed Vegfa expression levels in Foxp1cKO neurons at E12.5. This analysis demonstrated no significant difference in Vegfa levels in the cortical plate in the absence of Foxp1. Later changes may be observed, but we feel confident that we can state that the phenotypes we see at E12.5 and E13.5 are due to early loss of Foxp1 in RG.

“2. Can inhibition of HIF1a signaling (or VEGFa) in RG diminish or abolish the phenotype in Foxp1^{ff};EMX1-Cre mice?”

To address this concern, we have performed *in vitro* experiments to determine whether promoting the stabilization of HIF-1 α can rescue the progenitor phenotype we observe in Foxp1 deficient NPCs we observe both *in vivo* and *in vitro* (see revised Figures 4 and EV4). This analysis has demonstrated that pharmacological stabilization of HIF-1 α is sufficient to rescue the progenitor phenotype we see in Foxp1KO NPCs, restoring progenitor numbers to control levels.

“Other concerns:

1. Fig. 1. Please include the quantification data of Foxp1 and TUJ1 expression levels during embryonic cortical brain development (E12.5 to E14.5) to demonstrate their relationships with blood vessel development; How about other blood vessel markers (e.g., CD31), in addition to IB4?”

To address this concern, we have included Foxp1 expression levels as requested. Rather than measure TUJ1 expression levels we measured the area of the cortical plate between E12.5-14.5. To demonstrate the switch to asymmetric neurogenic divisions at E13.5, we have included counts of the number of Tbr2⁺ intermediate progenitors at each timepoint.

We have found that IB4 exclusively co-labels CD31⁺ endothelial cells at the timepoints we analyze in this study and does not appear to label other cell types. We have used IB4 in our analyses as the signal is clearer compared to antibody labeling such as for CD31. We have illustrated this in Appendix Figure 1.

“2. Fig. 2. In addition to the IHC, please verify the Foxp1, HIF1a, and its target genes' expression by western blots.”

We have provided western blot analyses of HIF-1 α , Glut1, Ldha and Vegfa at E11.5, E12.5 and E13.5. These demonstrate the temporal downregulation of HIF-1 α and the corresponding increase in Glut1 protein. These analyses demonstrate that Ldha and Vegfa levels appear to decrease as HIF-1 α decreases but remain stable between E12.5 and E13.5. These results highlight the sometimes counter-intuitive nature of HIF-1a signaling. They also come with the caveat that the lysates collected are not homogeneous and include cell types other than RG, including endothelial cells.

“3. Fig. 3. Again, additional methods (e.g., western blots) are necessary to verify the expression levels of HIF1a's target genes.”

Given the heterogeneous nature of the cell types represented in cell lysates from control and mutant cortices, and the changes in the composition of cell types in control and Foxp1cKO cortices, we believe IHC analysis of protein levels with an RG-specific marker is the most appropriate way to measure changes in protein expression.

“4. Fig. 4. As IB4 labels additional cell types in addition to blood vessels, it is necessary to immunostain other markers for blood vessels.”

The reviewer correctly notes that IB4 has the capacity to label other cell types. However, in the analyses shown in Figure 4 (which have been relocated to Figure 3 in our revised submission), we restrict our measurements to contiguous blood vessels using IB4 to define these structures and, therefore, other cell types that might be labeled by the antibody are not included in this analysis. In addition, we have added images of CD31 and IB4 co-staining at E12.5 and E14.5. This analysis demonstrates that at these stages of neurogenesis, IB4-labeled cells appear to be exclusively CD31⁺ (see Appendix Figure 1). However, the images generated with the IB4 antibody were clearer and enabled us to better visualize features such as filopodia.

“Referee #3:

The manuscript by Buth et al, identify a role for the transcription factor FoxP1 in the regulation of cortical angiogenesis during neurogenesis via the regulation of HIF1alpha and VEGFA signaling in neural progenitors.

The role of FoxP1 in regulating the switch between early and late cortical neurons progenitors has been previously established by the authors. The current manuscript nicely demonstrates the role of FoxP1 in transcriptome profiles modification associated with this switch. In particular, the description of genes and regulators of cell metabolism and growth as well as the communication with the extracellular environment is physiologically and pathologically relevant.

The manuscript is well written. Data mostly supports the conclusion, and the regulation of target genes expression, in particular HIF-1alpha and VEGFA, in neural progenitors in response to FoxP1 activity is convincing. The main concern regarding this study is the lack of functional evidence between the expression of VEGF-A in neural progenitors and the increase angiogenesis in the absence of FoxP1 activity. Many other cell types could potentially express VEGFA and be impacted in a non-cell autonomous manner by the depletion of FoxP1. To clarify, while the description of the phenotype supports the hypothesis of a potential functional relevance of VEGFA derived from late-neural progenitors in the angiogenic process during cortical development, it does not demonstrate it.”

We have analyzed the expression levels of Vegfa in control and Foxp1^{ckO} neurons at E12.5, the time at which we first detect a vascularization phenotype (see Expanded View Figure EV3 in the revised manuscript). This analysis demonstrated that there is no effect on Vegfa levels in neurons in the absence of Foxp1. Therefore, while we cannot rule out an impact on vascularization by loss of Foxp1 in other cell types at later time points than we have analyzed, our results are most consistent with the changes occurring due to the loss of Vegfa produced by RG.

“1-The authors mentioned (page 3, line 73): "Nevertheless, the developmental factors that coordinate angiogenesis with neurogenesis remain poorly defined".

This statement is not accurate. Previous publications reported the role of VEGFA during neurogenesis and angiogenesis coordination. Some of them associated VEGFA expression from neural progenitors, neurons and/or astrocytes with coupling between neurogenesis and angiogenesis. While astrocytes are not relevant in this study because of the developmental timing, newborn neurons or other cell types could be considered.”

We thank the reviewer for this comment and apologize for any confusion. Our point was that there are few studies that have defined how Vegfa is regulated, i.e. the intrinsic factors that act upstream of Vegfa/vascularization to coordinate the timing of neurogenesis and angiogenesis. However, to avoid confusion to readers, we have removed this problematic sentence from the text.

“2-Fig2T-W and X. While I am open to believe it. It is not clear with the pictures and magnification provided that the expression VEGFA colocalizes mainly with the Nestin+ cells.”

We have added high-magnification images that we hope better to illustrate the expression of Vegfa in Nestin⁺ RG (see Expanded View Figure EV3 in the revised submission).

“Opening the possibility for other cell populations expressing VEGFA at later stages.

Similar observation for Fig. 3 data quantification. For fig2 and 3, if my understanding is correct, intensity of the staining is quantified in the region of interest (not specifically in nestin⁺ cells). Would that be possible to quantify expression (mRNA and/or protein) specifically in neural progenitors?”

We apologize for any confusion. Our analysis of Vegfa levels is indeed restricted to Nestin⁺ cells within the ventricular zone. We have improved our description in the text to clarify this point.

“3-VEGF-A has been shown to be expressed in different types of neurons, including newborn cortical neurons. If depletion of FoxP1 promotes the neurogenic division of neural progenitors. How the increase angiogenic phenotype observed after depletion of FoxP1 is a direct consequence of VEGFA expression in neural progenitors or an indirect result of the increase in neuronal VEGF-A expression because an increased number of differentiated neurons is unclear. Neuronal VEGF-A expression should be analyzed in the different conditions of FoxP1 expression.”

We thank the reviewer for raising this concern. We see increased vascularization of the Foxp1^{CKO} cortex at E12.5 prior to the increase in neurons at E13.5 (Pearson et al., Cell Reports, 2020). Additionally, we have analyzed the expression levels of Vegfa in Foxp1^{CKO} cortical plate neurons and demonstrated that Vegfa levels do not appear to be affected by Foxp1 loss (see Expanded View Figure 3EV). Furthermore, we have performed in vitro experiments that show that when we treat Foxp1 mutant NPCs with a drug that stabilizes HIF-1 α protein prior to differentiation, we can rescue their premature differentiation phenotype. While these data do not rule out an effect at later time points, they show that neuronal Vegfa levels are not impacted by loss of Foxp1.

We have included an analysis of Vegfa and Foxp1 expression in neurons at later time points. This analysis demonstrates that while we see co-expression in neurons, we do not see the reciprocal relationship that exists in RG (see Figure 3EV). Thus, at the stages we focus on, we have no reason to believe that changes in Vegfa in neurons contribute to the vascularization phenotype.

“4-The description of VEGFA expression and regulation in neural progenitors is not completely novel. However, the concept of a switch in VEGFA expression associated with a late type of neural progenitors committed to neuronal differentiation is interesting. Conditional VEGFA expression in neural progenitors before the switch from early to late neural progenitors could demonstrate the role of VEGF-A expression in progenitors during cortical angiogenesis.”

“5-The contribution of VEGF-A derived from late-neural progenitors could also be functionally investigated. Conditional

deletion of VEGF-A in neural progenitors derived from radial glia will help draw a more definitive conclusion regarding the role of VEGFA expression in late neural progenitors in the coordination between neurogenesis and angiogenesis.”

While we agree with the reviewer that these are interesting and important avenues of investigation, we feel that a comprehensive exploration of the effects of Vegfa signaling and contributions in primary vs secondary progenitors goes beyond the scope of our current study. Our intent here was to broadly characterize the functions of Foxp1 in early radial glia focusing on its impact on HIF-1 α signaling, angiogenesis, and neurogenesis. We believe that establishing this link alone constitutes a significant contribution and opens the door to new lines of future investigation to be pursued, including determining the actions of Vegfa in a fine-grained manner as suggested.

“6-Foxp1 is a target of microRNA-9. miR-9 has been shown as coupling neurogenesis and angiogenesis by the inhibition of factors regulating VEGFA expression. Potential miR-9 expression in radial glia, early and late progenitors in the cortex, as well as a putative miR-9 dependent regulation of FoxP1 during early to late progenitor identity switch could be discussed.”

We thank the reviewer for bringing this to our attention, and we have accordingly incorporated this idea in our revised discussion.

“7-The title of the manuscript indicates that FoxP1 suppresses cortical angiogenesis. The data supports this statement. But Foxp1 has also been associated with promotion of the angiogenesis process. While a context dependent function of FoxP1 could explain this difference, this should be mentioned and discussed by the authors.”

Again, we thank the reviewer for raising this interesting point and have made mention of it in the last paragraph of our revised discussion.

Dear Caroline,

Thank you for the submission of your revised manuscript. We have now received the enclosed reports from the referees, as well as cross-comments. While referees 3 and 4 are overall positive, referees 2 and 3 still have suggestions for how the study could be further strengthened and developed. However, and considering the cross-comments, we can offer to publish your manuscript without further experimentation.

Please address the last referee comments in a point-by-point response. You are of course also very welcome to add more experiments along the lines suggested by the referees, also in the cross-comments. It would be good to replace "regulate" in the ms title, may be with "promote"?

A few editorial requests will also need to be addressed before we can proceed with the official acceptance of your manuscript:

- Please upload the final ms text and the figures separately. All figures need to be upload in portrait format.
- Please add up to 5 keywords to the ms file.
- The "Data Availability Section" needs to be named as such.
- The conflict of interest subheading needs to be corrected to "Disclosure and Competing Interest Statement"
- Please correct names M. Elizabeth Ross in the ms file versus Margaret Ross in our online ms submission system.
- Please correct the reference format to the EMBO reports (Harvard) style. "et al" needs to be used after 10 author names.
- Please upload a completed author checklist with your final ms. The checklist can be downloaded here:
<<https://www.embopress.org/page/journal/14693178/authorguide>>
- Some FUNDING INFO is missing in our online system, and some is missing in the ms file. Please correct.
- Please upload Tables EV1-EV4 as individual files.
- The Appendix figures need an "S" in their name: Appendix Figure S1, etc. Please correct.
- The manuscript sections should be in the following order: Title page - Abstract & Keywords - Introduction - Results & Discussion - Materials & Methods - Data Availability - Acknowledgments - Disclosure Statement & Competing Interests - References - Figure Legends - Expanded View Figure Legends.
- Please address these comments from our data editors:
 1. Please note that a separate 'Data Information' section is required in the legends of figures 1d-f; 2a-l, o-u, 3a-d, g-l; 4c-e; EV 2a-x; EV 3a-u, EV 4a-f, h-p.
 2. Please note that the scale bar information in the legend for figures EV 4h-p is incorrectly labelled as EV 4g-o. This needs to be rectified.
 3. Please indicate the statistical test used for data analysis in the legends of figure 1b; EV 1a-b.
 4. Please note that information related to n is missing in the legend of figure 1c.
 5. Please note that the error bars are not defined in the legends of figures 1c, k.
 6. Please note that the white arrow is not defined in the legends of figures 1e-f. This needs to be rectified.
 7. Please note that the arrowheads are not defined in the legends of figures 3g-h. This needs to be rectified,

EMBO press papers are accompanied online by A) a short (1-2 sentences) summary of the findings and their significance, B) 2-3 bullet points highlighting key results and C) a synopsis image that is exactly 550 pixels wide and 200-600 pixels high (the height is variable). You can either show a model or key data in the synopsis image. Please note that text needs to be readable at the final size. Please send us this information along with the final manuscript.

Esther Schnapp, PhD

Referee #2:

The revised manuscript addressed some of the concerns raised previously. However, some key questions remain not to be addressed. For example, the question regarding evidence for RGC specific Foxp1's function in angiogenesis, the authors only provided evidence for their potential association. It would be more convincing if the authors could show that specifically KO Foxp1 in RGC cells by use of a more specific Cre line, but not Emx1-Cre, could increase the angiogenesis. EMX1-Cre mouse line is known to express Cre in both excitatory neurons and glia [astrocytes and radial glia (RG)] in both cortex and hippocampus. Allen's RNAseq data base suggests that Foxp1 is expressed in both excitatory neurons and glial cells. The IHC data in Fig. 2 also showed that Foxp1 was not only enriched in the VZ, but also expressed in neurons. Thus, the conclusion of "early loss of Foxp1 in RG-radial glia", based on the data from Foxp1^{f/f};EMX1-Cre embryos (at E12.5), needs additional evidence. How about the phenotypes in Floxp1 CKO in embryonic neurons?

Referee #3:

In this revised version of the manuscript by Buth et al, the authors have addressed many of my previous concerns.

They provided an elegant in vitro approach using Foxp1KO spheroids. This new set of experiments strongly supports the cell autonomous role of Foxp1 in the regulation of HIF-1 signaling and associated neurogenesis defects. It is a very interesting, maybe unexpected, finding demonstrating that HIF1a stabilization rescues the neurogenic phenotype in vitro, in absence of vascularization.

However, this new dataset does not provide any additional evidence on the role of Foxp1 regarding the vascular phenotype. While one of the main conclusions of the study is about the role Foxp1 expression in radial to regulate angiogenesis, this phenotype has not been investigated in vitro. I understand the complexity of doing it in this in vitro system, but maybe the authors could have tried to rescue the angiogenic phenotype by inhibition of HIF1 or vegfa signaling in vivo to draw a more causal link between RG-derived vegf expression and angiogenesis defects.

Minor comment.

In their response, the authors mentioned:

"Therefore, while we cannot rule out an impact on vascularization by loss of Foxp1 in other cell types at later time points than we have analyzed, our results are most consistent with the changes occurring due to the loss of Vegfa produced by RG".

If my understanding of their work is correct; I believe that they mean increase VEGF produced by RG.

Referee #4:

The revised manuscript by Buth, Dyeovich et al. addresses the main concern raised in the original manuscript and the authors provide persuasive and consistent responses to all points. They include an alternative in vitro approach that recapitulates the in vivo results and demonstrates a cell-autonomous role of Foxp1 in HIF-1 signaling pathway, providing a link between HIF-1 signaling and neural progenitor behaviour. More, these results corroborate the influence of neural progenitor Foxp1 on cortical angiogenesis.

Comprehensive qPCR measurements, rearrangements of the figures, and refining of the text to accurately depict the findings greatly enhance the current version of the manuscript.

Therefore, I recommend publication in EMBO Reports.

Remarks

The manuscript would need small adjustments:

- Figure Legend EV4, Line 566: 50 μ m scale bar refers to panels H-P and not G-O.
- Figure 2EV should match the flow of the "results and discussion" section (lines 148-164), namely panels T-X should precede panels H-S. I suggest rearranging the order of the panels.

Cross-comments from referee 3:

I agree with reviewer2 comments.

I don't agree with the following comment from reviewer4. "More, these results corroborate the influence of neural progenitor

Foxp1 on cortical angiogenesis".

I think that the recently added in vitro experiments are only speculative on the role regulating angiogenesis.

It is a good study with some weakness. I guess it is about how good the correlations are vs. how small the doubt is.

Currently the study is not completely ruling out that another cellular component (in addition of RGC), neurons?, participates in the angiogenic phenotype.

This being said I don't know if all the points should be addressed. It would be a tremendous amount of work. but inhibition of HIF1 or vegfA in vivo will be a great addition. But it will not fix the reviewer 2 concerns.

Cross-comments from referee 4:

The reviewers still have concerns regarding the role of neural Foxp1 on the vascular phenotype and they suggest providing evidence in vivo either by modulating HIF-1a or VEGF expression or using alternative genetic models.

I understand this concern, which we also brought up in the first revision. We suggested similar in vivo experiments along with an in vitro approach to rule out a vessel-mediated effect on Foxp1 KO neural progenitor behaviour. The in vitro approach they provide recapitulates the phenotype seen in vivo, the cell-autonomous role of Foxp1 in the regulation of HIF-1a signaling and neural progenitor fate decision, and strengthens the potential influence of neural FoxP1 on angiogenesis by showing increased HIF1-a target genes, such as Vegfa, in Foxp1 cKO spheroids. The influence of Vegfa on angiogenesis has been already described (Lange et al., 2016; Gerhardt et al., 2003; Ruhrberg et al., 2002) and it is known that neural progenitors are the main source of Vegfa during brain development (Haigh et al., 2003; Raab et al., 2004; Di Marco et al., 2020). Therefore, the authors can come to the conclusion that Foxp1 regulates HIF1a target gene Vegfa with a consequent impact on cortical angiogenesis. Additional experiments, such as Vegfa in vivo modulation would be necessary to provide a more causal link between RG-derived VEGF expression and angiogenesis defects- as referee #3 suggested. This would enhance the impact of the study. While we suggested in-utero electroporation to reduce Vegfa expression in Foxp1 cKO embryos, the authors found this alternative technically challenging (answer to our second suggestion under point 1). They could still consider modulating HIF1a in Foxp1 cKO in vivo by exposing pregnant dams to decreased oxygen levels, as in Lange et al 2016.

Regarding the RGC specific Foxp1's function in angiogenesis raised by referee #2, I agree more specific Cre lines are available and the authors should specifically downregulate Foxp1 in RGC cells in support of their claim. However, switching the genetic model and performing again the experiments with a different genetic line would be too demanding. To address this concern, the authors provide a comprehensive answer supported by RNAScope and IHC data. Despite Foxp1 is expressed in RG and neurons and both populations could affect angiogenesis, the authors rule out a neuron-mediated effect by showing Vegfa expression in these populations. Foxp1 cKO shows significant upregulation of Vegfa in the apical ventricular zone (VZ), where RG cells line, while there is no difference in Vegf expression in neurons (Figure 2, 3EV). Given RG cells are the major source of Vegfa at the time of the analysis, the Vegf upregulation exclusively in the Foxp1 cKO VZ supports a RG Foxp1's function at this embryonic stage. Later - as they have already stated- changes may be observed.

Referee #2:

The revised manuscript addressed some of the concerns raised previously. However, some key questions remain not to be addressed. For example, the question regarding evidence for RGC specific Foxp1's function in angiogenesis, the authors only provided evidence for their potential association. It would be more convincing if the authors could show that specifically KO Foxp1 in RGC cells by use of a more specific Cre line, but not Emx1-Cre, could increase the angiogenesis.

EMX1-Cre mouse line is known to express Cre in both excitatory neurons and glia [astrocytes and radial glia (RG)] in both cortex and hippocampus. Allen's RNAseq data base suggests that Foxp1 is expressed in both excitatory neurons and glial cells.

The IHC data in Fig. 2 also showed that Foxp1 was not only enriched in the VZ, but also expressed in neurons. Thus, the conclusion of "early loss of Foxp1 in RG-radial glia", based on the data from Foxp1^{f/f};EMX1-Cre embryos (at E12.5), needs additional evidence. How about the phenotypes in Foxp1 CKO in embryonic neurons?

Our revised manuscript aims to determine how Foxp1 regulates neural progenitor maintenance. We show that Foxp1 acts upstream of HIF1 alpha to attenuate the downstream effectors and maintain progenitor maintenance. Our in vitro studies have demonstrated that this is a cell autonomous effect, and as referee 4 says, we have provided a link between Foxp1, HIF1a signaling and progenitor behavior. Also, our data corroborates previous studies that show that neural progenitors influence angiogenesis.

Our revised manuscript provides data that speaks directly to their concerns that elevations in Vegfa could be influenced by Foxp1 loss in neurons rather than progenitors. We have shown:

- In EV Figure 3A-C, we show that whereas Foxp1 and Vegfa are generally reciprocal in their expression in the ventricular zone, with increases in Vegfa occurring as Foxp1 levels decline across time in development, Vegfa expression in the cortical plate appears coincident with Foxp1.
- Moreover, in Figure EV3T-V, we demonstrate that Foxp1 levels in the cortical plate are not significantly changed in control vs. Foxp1 mutant cortices (compare quantitation in EV 3V to the same measurement in the ventricular zone shown in Figures 2T and 2X).
- We show that angiogenesis proceeds the switch to neurogenesis (Figure 1D-F) and prior to the expression of Vegfa in the cortex (either in radial glia or neurons). Also, we show that the earliest angiogenesis phenotype we see precedes neurogenesis (Figure 1D, Figure 3A-D).

We understand the reviewer's concerns regarding the Emx1Cre. However, to the best of our knowledge, there is not a more specific RG-Cre to be used. What's more, the effects of any RG-specific Cre line with respect to gene deletion will be retained in all RG progeny, and therefore affect both neurons and glia. Our studies are limited to time points at which a glial contribution would not be relevant. Additionally, we have shown that Vegfa levels are not impacted

by loss of Foxp1 in neurons (Figure EV3T-V). We cannot rule out an influence on angiogenesis at later time points and have added a sentence to our discussion to reflect this.

Referee #3:

In this revised version of the manuscript by Buth et al, the authors have addressed many of my previous concerns.

They provided an elegant in vitro approach using Foxp1KO spheroids. This new set of experiments strongly supports the cell autonomous role of Foxp1 in the regulation of HIF-1 signaling and associated neurogenesis defects. It is a very interesting, maybe unexpected, finding demonstrating that HIF1a stabilization rescues the neurogenic phenotype in vitro, in absence of vascularization.

However, this new dataset does not provide any additional evidence on the role of Foxp1 regarding the vascular phenotype. While one of the main conclusions of the study is about the role Foxp1 expression in radial to regulate angiogenesis, this phenotype has not been investigated in vitro. I understand the complexity of doing it in this in vitro system, but maybe the authors could have tried to rescue the angiogenic phenotype by inhibition of HIF1 or vegfa signaling in vivo to draw a more causal link between RG-derived vegf expression and angiogenesis defects.

We appreciate this suggestion. There are significant limitations to doing these experiments in vivo. For instance, the time point at which we would have to perform in utero electroporations would be prior to E13.5, which would significantly decrease the viability of electroporated embryos. Alternatively, pharmacological inhibition of Vegfa or HIF1a would not specifically target expression in radial glia.

Minor comment.

In their response, the authors mentioned:

"Therefore, while we cannot rule out an impact on vascularization by loss of Foxp1 in other cell types at later time points than we have analyzed, our results are most consistent with the changes occurring due to the loss of Vegfa produced by RG".

If my understanding of their work is correct; I believe that they mean increase VEGF produced by RG.

Yes, apologies, we mean increased VEGFA.

Referee #4:

The revised manuscript by Buth, Dyeovich et al. addresses the main concern raised in the original

manuscript and the authors provide persuasive and consistent responses to all points. They include an alternative in vitro approach that recapitulates the in vivo results and demonstrates a cell-autonomous role of Foxp1 in HIF-1 α signaling pathway, providing a link between HIF-1 α signaling and neural progenitor behaviour. More, these results corroborate the influence of neural progenitor Foxp1 on cortical angiogenesis.

Comprehensive qPCR measurements, rearrangements of the figures, and refining of the text to accurately depict the findings greatly enhance the current version of the manuscript.

Therefore, I recommend publication in EMBO Reports.

Remarks

The manuscript would need small adjustments:

- Figure Legend EV4, Line 566: 50 μ m scale bar refers to panels H-P and not G-O.

We have made this correction.

- Figure 2EV should match the flow of the "results and discussion" section (lines 148-164), namely panels T-X should precede panels H-S. I suggest rearranging the order of the panels.

We have made this adjustment.

Dr. Caroline Pearson
Weill Cornell Medicine
413 69th street east
Belfer Building
New York, NY 10021
United States

Dear Caroline,

I am very pleased to accept your manuscript for publication in the next available issue of EMBO reports. Thank you for your contribution to our journal.

Best wishes,
Esther
